# MotionDirector: Motion Customization of Text-to-Video Diffusion Models

## Abstract

Large-scale pre-trained diffusion models have exhibited remarkable capabilities in diverse video generations. Given a set of video clips of the same motion concept, the task of Motion Customization is to adapt existing text-to-video diffusion models to generate videos with this motion. For example, generating a video with a car moving in a prescribed manner under specific camera movements to make a movie, or a video illustrating how a bear would lift weights to inspire creators. Adaptation methods have been developed for customizing appearance like subject or style, yet unexplored for motion. It is straightforward to extend mainstream adaption methods for motion customization, including full model tuning, parameter-efficient tuning of additional layers, and Low-Rank Adaptions (LoRAs). However, the motion concept learned by these methods is often coupled with the limited appearances in the training videos, making it difficult to generalize the customized motion to other appearances. To overcome this challenge, we propose MotionDirector, with a dual-path LoRAs architecture to decouple the learning of appearance and motion. Further, we design a novel appearance-debiased temporal loss to mitigate the influence of appearance on the temporal training objective. Experimental results show the proposed method can generate videos of diverse appearances for the customized motions. Our method also supports various downstream applications, such as the mixing of different videos with their appearance and motion respectively, and animating a single image with customized motions. Our code and model weights will be released.

## 1 Introduction

Text-to-video diffusion models (Ho et al., 2022; Singer et al., 2022; He et al., 2022) are approaching generating high-quality diverse videos given text instructions. The open-sourcing of foundational text-to-video models (Wang et al., 2023a; Sterling, 2023) pre-trained on large-scale data has sparked enthusiasm for video generation in both the community and academia. Users can create videos that are either realistic or imaginative simply by providing text prompts. While foundation models generate diverse videos from the same text, adapting them to generate more specific content can better accommodate the preferences of users. Similar to the customization of text-to-image foundation models (Ruiz et al., 2023), tuning video foundation models to generate videos of a certain concept of appearance, like subject or style, has also been explored (He et al., 2022). Compared with images, videos consist of not just appearances but also motion dynamics. Users may desire to create videos with specific motions, such as a car moving forward and then turning left under a predefined camera perspective, as illustrated on the right side in Fig. 1. However, customizing the motions in the text-to-video generation is still unexplored.

The task of Motion Customization is formulated as follows: given reference videos representing a motion concept, the objective is to turn the pre-trained foundation models into generating videos that exhibit this particular motion. In contrast, previous works on appearance customization adapt the foundation models to generate samples with desired appearance, like subject or style, given reference videos or images representing such appearance (Ruiz et al., 2023; He et al., 2022). It is straightforward to use previous adaption methods for motion customization. For example, on the given reference videos, fine-tuning the weights of foundation models (Ruiz et al., 2023), parameter-efficient tuning additional layers (Wu et al., 2022), or training Low-Rank Adaptions (LoRAs) (Hu et al., 2021) injected in the layers of foundation models. However, customizing diffusion models

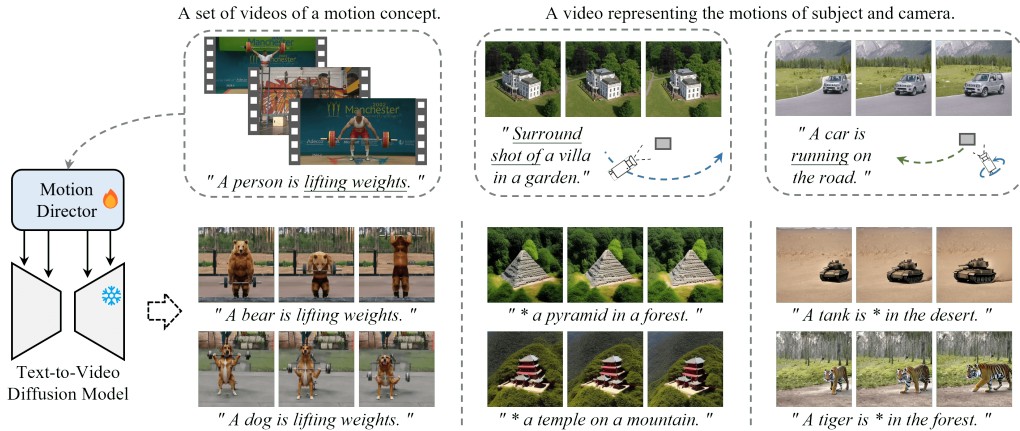

Figure 1: Motion customization of the text-to-video diffusion model.

to generate desired motions without harming their appearance diversity is challenging because the motion and appearance are coupled with each other at the step-by-step denoising stage. Directly deploying previous adaption methods to learn motions makes the models fit the limited appearances seen in the reference videos, posing challenges in generalizing the learned motions to various appearances. Recent works on controllable text-to-video generations (He et al., 2022; Esser et al., 2023; Wang et al., 2023b) generate videos controlled by signals representing pre-defined motions. However, the control signals, such as depth maps or edges, impose constraints on the shapes of subjects and backgrounds, thus influencing the appearance of generated videos in a coupled way. Besides, these methods accept only one sequence of control signals to generate one video, which may not be suitable for users seeking certain motion types without strict spatial constraints, such as the example of lifting weights in Fig. 1.

To achieve motion customization of text-to-video diffusion models while preserving appearance diversity, we propose the MotionDirector, which tunes the foundation models to learn the appearance and motions in the given single or multiple reference videos in a decoupled way. MotionDirector tunes the models with low-rank adaptions (LoRAs) while keeping their pre-trained parameters fixed to retain the learned generation knowledge. Specifically, the MotionDirector employs a dual-path architecture, as shown in Fig. 3. For each video, a spatial path consists of a foundation model with trainable spatial LoRAs injected into its spatial transformer layers. These spatial LoRAs are trained on a single frame randomly sampled per training step to capture the appearance characteristics of the input videos. The temporal path, on the other hand, is a replica of the foundation model that shares the spatial LoRAs with the spatial path to fit the appearance of the corresponding input video. Additionally, the temporal transformers in this path are equipped with temporal LoRAs, which are trained on multiple frames of input videos to capture the underlying motion patterns. To further enhance the learning of motions, we propose an appearance-debiased temporal loss to mitigate the influence of appearance on the temporal training objective.

Only deploying the trained temporal LoRAs enables the foundation model to generate videos of the learned motions with diverse appearances, as shown in the second row of Fig 2. The decoupled paradigm further makes an interesting kind of video generation feasible, which is the mix of the appearance from one video with the motion from another video, called the mix of videos, as shown in the third row of Fig 2. The key to this success lies in that MotionDirector can decouple the appearance and motion of videos and then combine them from various source videos. It is achieved by injecting spatial LoRAs trained on one video and temporal LoRAs trained on another video into the foundation model. Besides, the learned motions can be deployed to animate images, as images can be treated as appearance providers, as shown in the last row of Fig 2.

We conducted experiments on two benchmarks with 86 different motions and over 600 text prompts to test proposed methods, baselines, and comparison methods. The results show our method can be applied to different diffusion-based foundation models and achieve motion customization of various motion concepts. On the UCF Sports Action benchmark, which includes 95 videos for 12 types

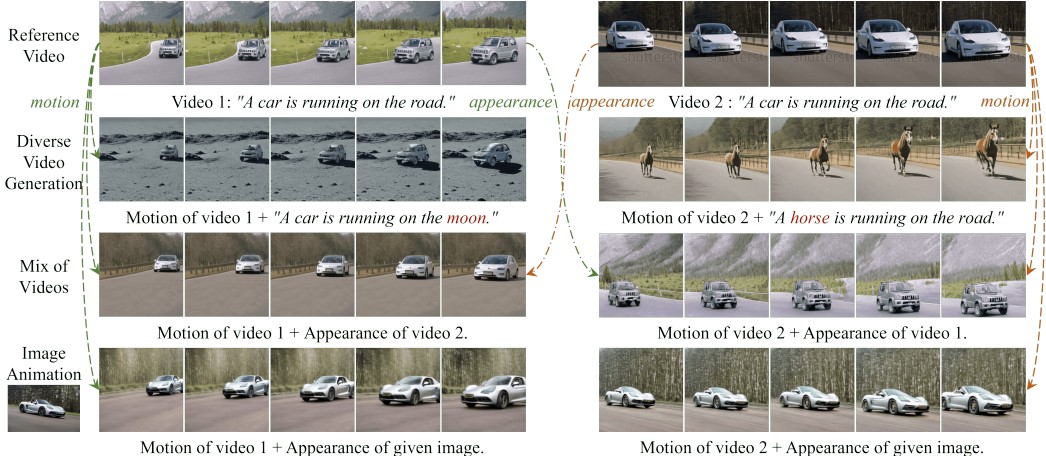

Figure 2: (Row 1) Take two videos to train the proposed MotionDirector, respectively. (Row 2) MotionDirector can generalize the learned motions to diverse appearances. (Row 3) MotionDirector can mix the learned motion and appearance from different videos to generate new videos. (Row 4) MotionDirector can animate a single image with learned motions.

of motion concepts and 72 labeled text prompts, human raters preferred MotionDirector for higher motion fidelity at least 75% of the time, significantly outperforming the 25% preferences of base models. On the LOVEU-TGVE-2023 benchmark, which includes 76 reference videos and 532 text prompts, MotionDirector outperforms controllable generation methods and the tuning-based method by a large margin, especially in the human preference for appearance diversity. Compared with these methods, our method avoids fitting the limited appearance of reference videos, and can generalize the learned motions to diverse appearances.

Our contributions are summarized as follows:

- We introduce and define the task of Motion Customization. The challenge lies in generalizing the customized motions to various appearances.

- We propose the MotionDirector with a dual-path architecture and a novel appearance-debiased temporal training objective, to decouple the learning of appearance and motion.

- Experiments on two benchmarks demonstrate that MotionDirector can customize various base models to generate diverse videos with desired motion concepts, and outperforms controllable generation methods and tuning-based methods.

## 2 RELATED WORK

**Text-to-Video Generation.** To achieve high-quality video generation, various methods have been developed, such as Generative Adversarial Networks (GANs) (Vondrick et al., 2016; Saito et al., 2017; Tulyakov et al., 2018; Balaji et al., 2019; Tian et al., 2020; Shen et al., 2023), autoregressive models (Srivastava et al., 2015; Yan et al., 2021; Le Moing et al., 2021; Hong et al., 2022; Ge et al., 2022) and implicit neural representations (Yu et al., 2021; Skorokhodov et al., 2021). Diffusion-based models (Ni et al., 2023; Yu et al., 2023; Mei & Patel, 2023; Voleti et al., 2022) are also approaching high-quality generation by training conditional 3D U-Nets to denoise from randomly sampled sequences of Gaussian noises. Recent foundation models (Ho et al., 2022; Singer et al., 2022; He et al., 2022; Luo et al., 2023; Blattmann et al., 2023; Zhang et al., 2023; Wang et al., 2023c) are pre-trained on large-scale image and video datasets (Schuhmann et al., 2022; Deng et al., 2009; Bain et al., 2021), to learn powerful generation ability. Some works turn text-to-image foundation models to text-to-video generation by manipulation on cross-frame attention or training additional temporal layers, like Tune-A-Video (Wu et al., 2022), Text2Video-Zero(Khachatryan et al., 2023), and AnimiteDiff (Guo et al., 2023). The recently open-sourced foundation models (Wang et al.,

2023a; Sterling, 2023) have ignited enthusiasm among users to generate realistic or imaginative videos, and further make it possible for users to customize and build their own private models.

**Generation Model Customization.** Customizing the pre-trained large foundation models can fit the preferences of users better while maintaining powerful generation knowledge without training from scratch. Previous customization methods for text-to-image diffusion models (Ruiz et al., 2023; Kumari et al., 2023; Gu et al., 2023; Chen et al., 2023b; Wei et al., 2023; Smith et al., 2023) aim to generate certain subjects or styles, given a set of example images. Dreambooth (Ruiz et al., 2023) or LoRA (Hu et al., 2021) can be simply applied to customizing video foundation models to generate videos with certain subjects or styles, given a set of reference video clips or images. The recently proposed VideoCrafter (He et al., 2023) has explored this, which we categorize as appearance customization. In addition to appearances, videos are also characterized by the motion dynamics of subjects and camera movements across frames. However, to the best of our knowledge, customizing the motions in generation for text-to-video diffusion models is still unexplored.

**Controllable Video Generation.** Controllable generation aims to ensure the generation results align with the given explicit control signals, such as depth maps, human pose, optical flows, etc. (Zhang & Agrawala, 2023; Zhao et al., 2023; Ma et al., 2023). For the controllable text-to-video generation methods, i.e. the VideoCrafter (He et al., 2022), VideoComposer (Wang et al., 2023b), Control-A-Video (Chen et al., 2023a), they train additional branches that take condition signals to align the generated videos with them. Unlike the human poses for specifically controlling the generation of human bodies, the general control singles, such as depth maps, are typically extracted from reference videos and are coupled with both appearance and motion. This results in the generation results being influenced by both the appearance and motion in reference videos. Applying these methods directly in motion customization is challenging when it comes to generalizing the desired motions to diverse appearances.

## 3 METHODOLOGY

### 3.1 PRELIMINARIES

**Video Diffusion Model.** Video diffusion models train a 3D U-Net to denoise from a randomly sampled sequence of Gaussian noises to generate videos, guided by text prompts. The 3D U-net basically consists of down-sample, middle, and up-sample blocks. Each block has several convolution layers, spatial transformers, and temporal transformers as shown in Fig 3. The 3D U-Net $\epsilon_\theta$ and a text encoder $\tau_\theta$ are jointly optimized by the noise-prediction loss, as detailed in (Dhariwal & Nichol, 2021):

$$\mathcal{L} = \mathbb{E}_{z_0, y, \epsilon \sim \mathcal{N}(0, I), t \sim \mathcal{U}(0, T)} \left[ \|\epsilon - \epsilon_\theta(z_t, t, \tau_\theta(y))\|_2^2 \right], \tag{1}$$

where $z_0$ is the latent code of the training videos, $y$ is the text prompt, $\epsilon$ is the Gaussian noise added to the latent code, and $t$ is the time step. As discussed in (Dhariwal & Nichol, 2021), the noised latent code $z_t$ is determined as:

$$z_t = \sqrt{\bar{\alpha}_t} z_0 + \sqrt{1 - \bar{\alpha}_t} \epsilon, \ \bar{\alpha}_t = \prod_{i=1}^{t} \alpha_t, \tag{2}$$

where $\alpha_t$ is a hyper-parameter controlling the noise strength.

**Low-Rank Adaption.** Low-rank adaption (LoRA) (Hu et al., 2021) was proposed to adapt the pre-trained large language models to downstream tasks. Recently it has been applied in text-to-image generation and text-to-video generation tasks to achieve appearance customization (Ryu, 2023; He et al., 2023). LoRA employs a low-rank factorization technique to update the weight matrix $W$ as

$$W = W_0 + \Delta W = W_0 + BA, \tag{3}$$

where $W_0 \in \mathbb{R}^{d \times k}$ represents the original weights of the pre-trained model, $B \in \mathbb{R}^{d \times r}$ and $A \in \mathbb{R}^{r \times k}$ represent the low-rank factors, where $r$ is much smaller than original dimensions $d$ and $k$. LoRA requires smaller computing sources than fine-tuning the weights of the entire network like DreamBooth (Ruiz et al., 2023), and it is convenient to spread and deploy as a plug-and-play plugin for pre-trained models.

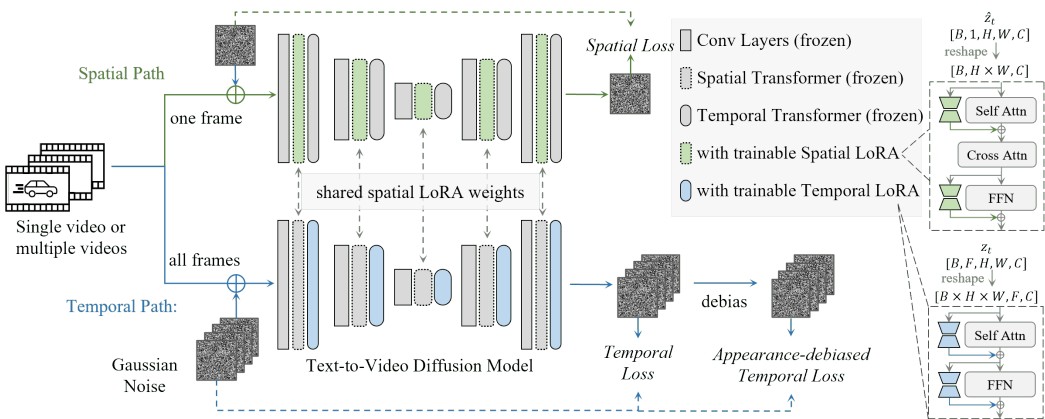

Figure 3: The dual-path architecture of the proposed method. All pre-trained weights of the base diffusion model remain fixed. In the spatial path, the spatial transformers are injected with trainable spatial LoRAs as shown on the right side. In the temporal path, the spatial transformers are injected with spatial LoRAs sharing weights with those ones in the spatial path, and the temporal transformers are injected with trainable temporal LoRAs.

## 3.2 DUAL-PATH LOW-RANK ADAPTIONS

At each time-step $t$, the 3D U-Net takes in the latent code $z_t \in \mathbb{R}^{b \times f \times w \times h \times c}$ and the conditional input $y$ (e.g., text), where $b$, $f$, $w$, $h$, $c$ represents the size of the batch, frame, width, height, and channel dimensions, respectively. The spatial transformers apply spatial self-attention along the spatial dimensions $w, h$ to improve the correlation between pixels, and then leverage the cross-attention between the latent code and the conditional input $y$ to improve textual alignment. The temporal transformers apply temporal self-attention along the frame dimension $f$ to improve the temporal consistency between frames. However, spatial and temporal information in the latent code gradually become coupled with each other during the step-by-step denoising stage. Attempting to directly learn and fit the motions in reference videos will inevitably lead to fitting their limited appearances. To address this problem, we propose to tune the spatial and temporal transformers in a dual-path way to learn the earn the appearance and motion in reference videos, respectively, as shown in Fig. 3. Specifically, for the spatial path, we inject LoRAs into spatial transformers to learn the appearance of training data, and for the temporal path, we inject LoRAs into temporal transformers to learn the motion in videos.

**Spatial LoRAs Training.** For the spatial path, we inject unique spatial LoRAs into the spatial transformers for each training video while keeping the weights of pre-trained 3D U-Net fixed. To maintain the learned strong and diverse textual alignment ability, we do not inject LoRAs into cross-attention layers of spatial transformers, since their weights influence the correlations between the pixels and text prompts. On the other hand, we inject LoRAs into spatial self-attention layers and feed-forward layers to update the correlations in spatial dimensions to enable the model to reconstruct the appearance of training data. For each training step, the spatial LoRAs are trained on a single frame randomly sampled from the training video to fit its appearance while ignoring its motion, based on spatial loss, which is reformulated as

$$\mathcal{L}_{spatial} = \mathbb{E}_{z_0,y,\epsilon,t,i \sim \mathcal{U}(0,F)} \left[ \|\epsilon - \epsilon_\theta(z_{t,i}, t, \tau_\theta(y))\|_2^2 \right], \quad (4)$$

where F is the number of frames of the training data and the $z_{t,i}$ is the sampled frame from the latent code $z_t$.

**Temporal LoRAs Training.** For the temporal path, we inject the temporal LoRAs into self-attention and feed-forward layers of temporal transformers to update the correlations along the frame dimension. Besides, the spatial transformers are injected with LoRAs sharing the same weights learned from the spatial path, to force the trainable temporal LoRAs to ignore the appearance of the training data. The temporal LoRAs could be simply trained on all frames of training data based on the temporal loss $\mathcal{L}_{org\text{-}temp}$, formulated in the same way as equation (1).

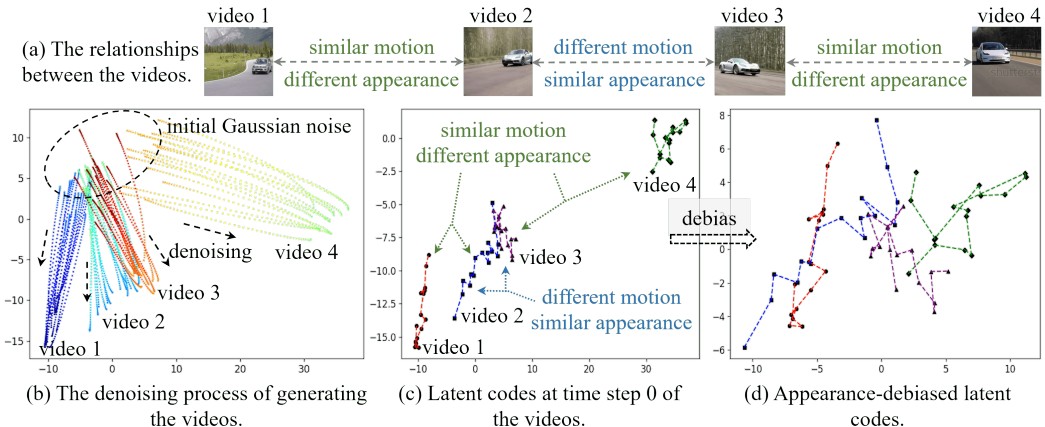

(a) The relationships between the videos.

(b) The denoising process of generating the videos.

(c) Latent codes at time step 0 of the videos.

(d) Appearance-debiased latent codes.

Figure 4: (a) Four example videos (the same as the videos in the first and fourth rows of Fig. 2) and their relationships in terms of motion and appearance. (b) We inverse the four videos based on the video diffusion model and visualize the denoising process. Each point corresponds to a latent code $z_{t,i,j}$ at time step $t$ of $i$-th frame of $j$-th video. (c) Take latent codes at time step 0 for example, the ones of the same video are connected in order of frames. We find that the internal connectivity structure between latent codes is more influenced by motion, while the distance between sets of latent codes is primarily affected by the difference in appearance. (d) The latent codes are debiased to eliminate the appearance bias among them while retaining their connectivity structure.

However, we notice that the noise prediction, in the temporal path, is still influenced by the appearance to some extent. As illustrated in Fig. 4, when considering the latent codes of each frame $z_{t,i}{}^{F}_{i=1}$ as a set of points in the latent space, motion primarily impacts the underlying dependencies between these point sets, whereas the distances between different sets of points are more influenced by appearance. To further decouple the motion from appearance, we proposed to eliminate the appearance bias among the noises and predicted noises, and calculate the appearance-debiased temporal loss on them. The debiasing of each noise $\epsilon_i \in \{\epsilon_i\}^{F}_{i=1}$ is as follows,

$$\phi(\epsilon_i) = \sqrt{\beta^2 + 1}\epsilon_i - \beta\epsilon_{anchor}, \qquad (5)$$

where $\beta$ is the strength factor controlling the debiasing strength and $\epsilon_{anchor}$ is the anchor among the frames from the same training data. In practice, we simply set $\beta = 1$ and randomly sample $\epsilon_i \in \{\epsilon_i\}^{F}_{i=1}$ as the anchor. The appearance-debiased temporal loss is reformulated as

$$\mathcal{L}_{ad\text{-}temp} = \mathbb{E}_{z_0,y,\epsilon,t}\left[\|\phi(\epsilon) - \phi(\epsilon_\theta(z_t, t, \tau_\theta(y)))\|^2_2\right]. \qquad (6)$$

For temporal LoRAs, the loss function is the combination of temporal loss and appearance-debiased temporal loss as follows,

$$\mathcal{L}_{temporal} = \mathcal{L}_{org\text{-}temp} + \mathcal{L}_{ad\text{-}temp}. \qquad (7)$$

**Motion Customization.** In the inference stage, we inject the trained temporal LoRAs into the pre-trained video diffusion model to enable it to generate diverse videos with the learned motion from the training data. If the training data is a single video, the learned motion will be a specific motion, such as an object first moving forward and then turning to the left. If the training data is a set of videos, the learned motion will be the motion concept provided by them, like lifting weights or playing golf. The motion concepts can be ones preferred by users or ones that lie in the long-tailed distribution that can not be synthesized well by pre-trained models. Since appearance and motion are decoupled by our method, the spatial LoRAs can also be used to influence the appearance of generated videos, as shown in Fig. 2. Users can flexibly adjust the influence strength of learned appearance and motion on the generation according to their preferences by simply setting the strength of LoRAs as $W = W_0 + \gamma\Delta W$, where $\gamma$ is called the LoRA scale, and $\Delta W$ is the learned weights.

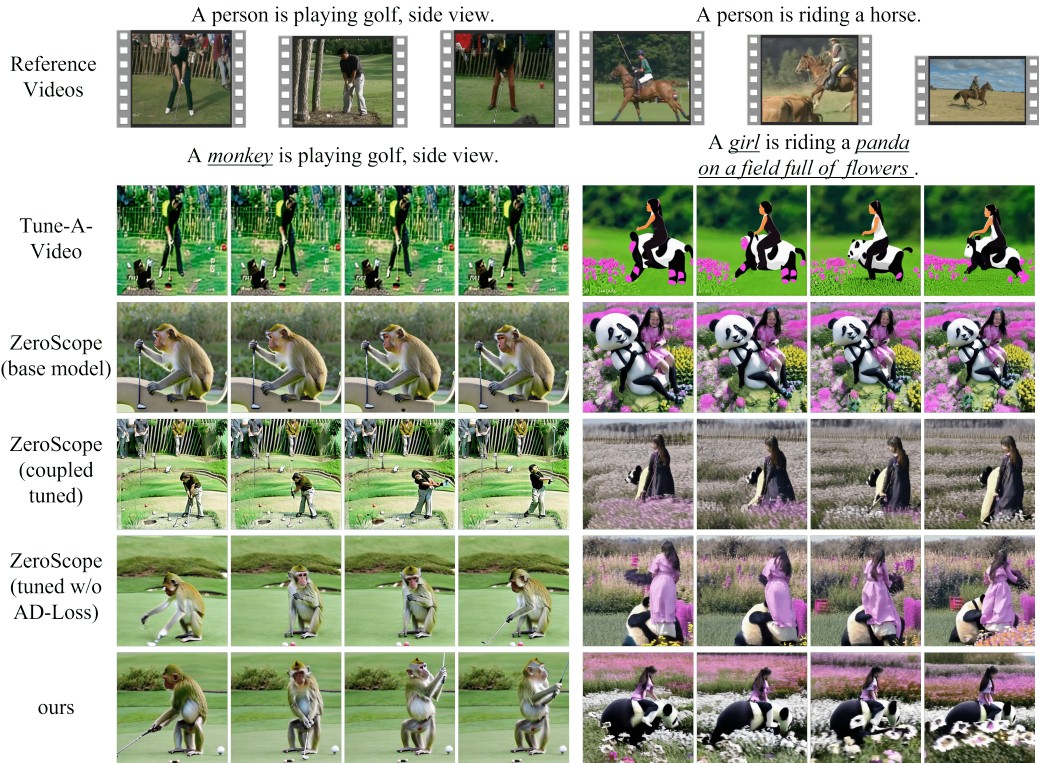

Figure 5: Qualitative comparison results of motion customization on multiple videos.

# 4 EXPERIMENTS

## 4.1 MOTION CUSTOMIZATION ON MULTIPLE VIDEOS

**Dataset.** We conduct experiments on the adapted UCF Sports Action data set (Soomro & Zamir, 2015), which includes 95 videos of 12 different human motions, like playing golf, lifting weights, etc. For each type of motion, we label one original text prompt describing the motion, such as "a person is playing golf, side view". For these motions, we set 72 different text prompts in total as input to generate videos using comparison methods, such as "a monkey is playing golf, side view".

**Comparison Methods.** We compare the proposed method with three baselines and the video generation method Tune-A-Video (Wu et al., 2022) that can be adapted to this task. Tune-A-Video was initially proposed for training temporal modules on a single video to learn its motion information, while here we adapt it to train on multiple videos. The baseline methods are compared with the proposed method on two different foundational text-to-video diffusion models, i.e. the ModelScope (Wang et al., 2023a) and the ZeroScope (Sterling, 2023). We employ three baseline methods: the first is directly applying the vanilla foundation models, the second is tuning the foundation models with LoRAs in a coupled manner, and the third is the proposed dual-path method excluding the appearance-debiased temporal loss.

**Qualitative Results** As shown in Fig. 5, taking a set of videos with motions of playing golf as training data, the Tune-A-Video fails to generate diverse appearances with the learned motions, like a monkey playing golf. To compare the baseline methods and proposed method fairly, we feed the same initial Gaussian noise to these methods to generate videos. The pre-trained foundation model, ZeroScope, correctly generates the appearance but lacks the realistic motion that swings a golf club, as those desired motions in the reference videos. The coupled tuned model could generate the desired motion but the learned motion is coupled with too much appearance information causing the generated subject in the video to be more like a human rather than a monkey. The last two rows show that the proposed dual-path LoRAs can avoid hurting the appearance generation and the

proposed appearance-debiased temporal loss enhances the learning of desired motion better. We could draw a similar conclusion from the second example showing the motion of riding a panda.

**Quantitative Results.** We evaluate the methods with automatic evaluations and human evaluations, and the results are shown in Table. 1.

*Automatic Metrics.* Following the LOVEU-TGVE competition (Wu et al., 2023), the appearance diversity is computing the average CLIP score (Hessel et al., 2021) between the diverse text prompts and all frames of the generated videos, the temporal consistency is the average CLIP score between frames, and the Pick Score is the average PickScore (Kirstain et al., 2023) between all frames of output videos.

*Human Preference.* On the Amazon MTurk [1], each generated video is evaluated by 5 human raters in terms of appearance diversity, temporal consistency, and motion fidelity, which evaluate whether the generated motion is similar to the references. To simplify the comparison for raters, they are asked to compare the results pairwise and select their preferred one, where the videos are shuffled and their source methods are anonymous. In Table. 1, the pairwise numbers "$p_1$ v.s. $p_2$" means $p_1\%$ results of the first method are preferred while $p_2\%$ results of the second method are preferred. Additional details are provided in the appendix (Sec. A.4).

The evaluation results show that coupled tuning will destroy the appearance diversity of pre-trained models, while our method will preserve it and achieve the highest motion fidelity.

Table 1: Automatic and human evaluations results of motion customization on multiple videos.

| | Automatic Evaluations | | | | Human Evaluations | | | |
|---|---|---|---|---|---|---|---|---|
| | Appearance Diversity (↑) | Temporal Consistency (↑) | Pick Score (↑) | | | Appearance Diversity | Temporal Consistency | Motion Fidelity |
| Tune-A-Video | 28.22 | 92.45 | 20.20 | v.s. Base Model (ModelScope) | | 25.00 v.s. 75.00 | 25.00 v.s. 75.00 | 40.00 v.s. 60.00 |
| | | | | v.s. Base Model (ZeroScope) | | 44.00 v.s. 56.00 | 16.67 v.s. 83.33 | 53.33 v.s. 46.67 |
| ModelScope Base Model | 28.55 | 92.54 | 20.33 | | | | | |
| Coupled Tuned | 25.66 (-2.89) | 90.66 | 19.85 | v.s. Base Model (ModelScope) | | **23.08 v.s. 76.92** | 40.00 v.s. 60.00 | 52.00 v.s. 48.00 |
| w/o AD-Loss | 28.32 (-0.23) | 91.17 | 20.34 | v.s. Base Model (ModelScope) | | 53.12 v.s. 46.88 | 49.84 v.s. 50.16 | 62.45 v.s. 37.55 |
| ours | **28.66 (+0.11)** | 92.36 | 20.59 | v.s. Base Model (ModelScope) | | **54.84 v.s. 45.16** | 56.00 v.s. 44.00 | **75.00 v.s. 25.00** |
| ZeroScope Base Model | 28.40 | 92.94 | 20.76 | | | | | |
| Coupled Tuned | 25.52 (-2.88) | 90.67 | 19.99 | v.s. Base Model (ZeroScope) | | **37.81 v.s. 62.19** | 41.67 v.s. 58.33 | 54.55 v.s. 45.45 |
| w/o AD-Loss | 28.61 (+0.21) | 91.37 | 20.56 | v.s. Base Model (ZeroScope) | | 50.10 v.s. 49.90 | 48.00 v.s. 52.00 | 58.33 v.s. 41.67 |
| ours | **28.94 (+0.54)** | 92.67 | 20.80 | v.s. Base Model (ZeroScope) | | **52.94 v.s. 47.06** | 55.00 v.s. 45.00 | **76.47 v.s. 23.53** |

## 4.2 MOTION CUSTOMIZATION ON A SINGLE VIDEO

**Dataset.** We conduct the comparison experiments on the open-sourced benchmark released by the LOVEU-TGVE competition at CVPR 2023 (Wu et al., 2023). The dataset comprises 76 videos, each originally associated with 4 editing text prompts. Additionally, we introduced 3 more prompts with significant changes.

**Comparison Methods.** We compare the proposed method with SOTA controllable generation methods, the VideoCrafter (He et al., 2022), VideoComposer (Wang et al., 2023b), and Control-A-Video (Chen et al., 2023a), and the tuning-based method Tune-A-Video(Wu et al., 2022). To ensure a fair comparison, we use the depth control mode of controllable generation methods, which is available in all of them.

Table 2: Automatic and human evaluations results of motion customization on a single video.

| | Automatic Evaluations | | | | Human Evaluations | | | | |
|---|---|---|---|---|---|---|---|---|---|
| | Text Alignment (↑) | Appearance Diversity (↑) | Temporal Consistency (↑) | Pick Score (↑) | | Text Alignment | Appearance Diversity | Temporal Consistency | Motion Fidelity |
| VideoComposer | 27.66 | 27.03 | 92.22 | 20.26 | ours v.s. VideoComposer | 54.55 v.s. 45.45 | **72.83 v.s. 27.17** | 61.57 v.s. 38.43 | 61.24 v.s. 38.76 |
| Control-a-Video | 26.54 | 25.35 | 92.63 | 19.75 | ours v.s. Control-A-Video | 68.00 v.s. 32.00 | **78.43 v.s. 21.57** | 71.28 v.s. 29.72 | 56.47 v.s. 43.53 |
| VideoCrafter | **28.03** | 27.69 | 92.26 | 20.12 | ours v.s. VideoCrafter | 52.72 v.s. 47.28 | **71.11 v.s. 28.89** | 60.22 v.s. 39.78 | 60.00 v.s. 40.00 |
| Tune-a-Video | 25.64 | 25.95 | 92.42 | 20.09 | ours v.s. Tune-A-Video | 67.86 v.s. 32.14 | **69.14 v.s. 30.86** | 71.67 v.s. 28.33 | 56.52 v.s. 43.48 |
| ours | 27.82 | **28.48** | **93.00** | **20.74** | | | | | |

**Qualitative and Quantitative Results.** As shown in Fig. 6, comparison methods fail to generalize the desired motions to diverse appearances, like the ears of bears and the Arc de Triomphe. In Table. 2, we refer to the alignment between the generated videos and the original 4 editing text prompts as text alignment, and the alignment with the 3 new text prompts with significant changes as appearance diversity. The results show that our method outperforms other methods by a large

---

[1] https://requester.mturk.com/

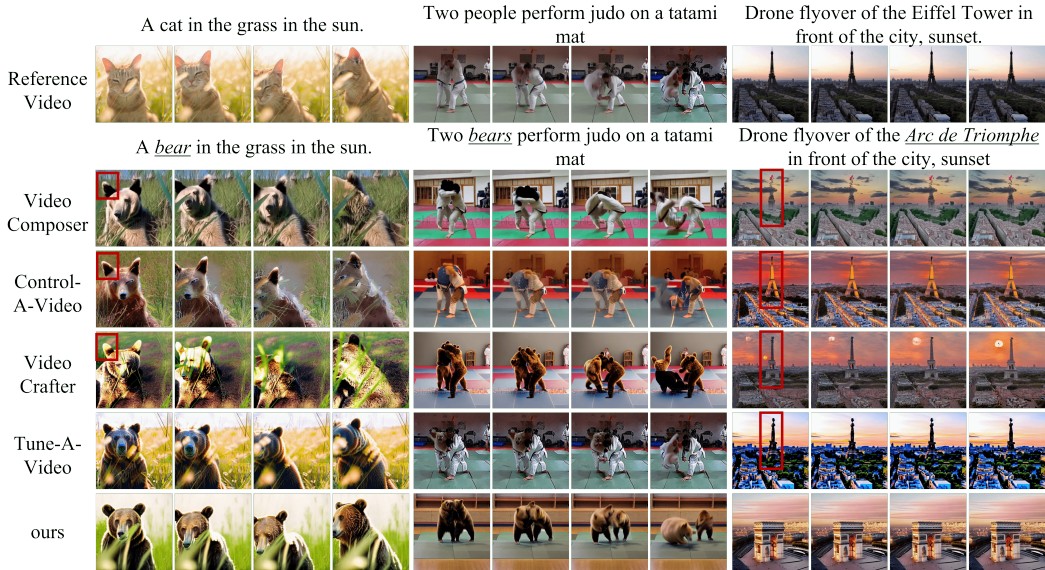

Figure 6: Qualitative comparison results of motion customization on a single video.

margin when generalizing the motions to diverse appearances, and achieves competitive motion fidelity.

### 4.3 EFFICIENCY PERFORMANCE

The lightweight LoRAs enable our method to tune the foundation models efficiently. Taking the foundation model ZeroScope for example, it has over 1.8 billion pre-trained parameters. Each set of trainable spatial and temporal LoRAs only adds 9 million and 12 million parameters, respectively. Requiring 14 GB VRAM, MotionDirector takes 20 minutes to converge on multiple reference videos, and 8 minutes for a single reference video, competitive to the 10 minutes required by Tune-A-Video (Wu et al., 2022). Additional details are provided in the appendix (Sec. A.2).

## 5 LIMITATIONS AND FUTURE WORKS

Despite the MotionDiector can learn the motions of one or two subjects in the reference videos, it is still hard to learn complex motions of multiple subjects, such as a group of boys playing soccer. Previous appearance customization methods suffer similar problems when generating multiple customized subjects (Gu et al., 2023). A possible solution is to further decouple the motions of different subjects in the latent space and learn them separately.

## 6 CONLCUSION

We introduce and formulate the task of Motion Customization, which is adapting the pre-trained foundation text-to-video diffusion models to generate videos with desired motions. The challenge of this task is generalizing the customized motions to various appearances. To overcome this challenge, we propose the MotionDirector with a dual-path architecture and a novel appearance-debiased temporal training objective to decouple the learning of appearance and motion. Experimental results show that MotionDirector can learn either desired motion concepts or specific motions of subjects and cameras, and generalize them to diverse appearances. The automatic and human evaluations on two benchmarks demonstrate the MontionDirector outperforms other methods in terms of appearance diversity and motion fidelity.

## 7 REPRODUCIBILITY STATEMENT

We make the following efforts to ensure the reproducibility of MotionDirector: (1) Our training and inference codes together with the trained model weights will be publicly available. (2) We provide training details in the appendix (Sec.A.2). (3) The reference videos in the two benchmarks are publicly accessible, and we will release the labeled text prompts. More details are provided in the appendix (Sec.A.3). (4) We provide the details of the human evaluation setups in the appendix (Sec.A.4).

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

# A APPENDIX

## A.1 COMPARISON OF TASKS

As shown in Fig. 7, we think video is characterized by two basic aspects, i.e. appearance and motion. Each video is represented as a point in Fig. 7, and a set of videos sharing similar motions or appearances lie in a close region. The task of appearance customization aims to learn the appearances, like the subjects or styles, in the reference videos, and generate new videos with the learned appearances, where the motions can be different from those in the reference videos. The task of controllable generation aims to generate videos aligned with the given control singles, such as a sequence of depth maps or edges. However, these control singles are often coupled with constraints on the appearances, limiting the appearance diversity of the generated videos. The task of motion customization aims to learn motions from a single video or multiple videos. If the motion is learned from a single video, then it will be a specific motion representing the movements of subjects and the camera, while if the motion is learned from multiple videos sharing the same motion concept, then it will be this motion concept, like how to play golf. The most important is that motion customization aims to generalize the learned motions to various appearances, and we can see a high appearance diversity in the generated results.

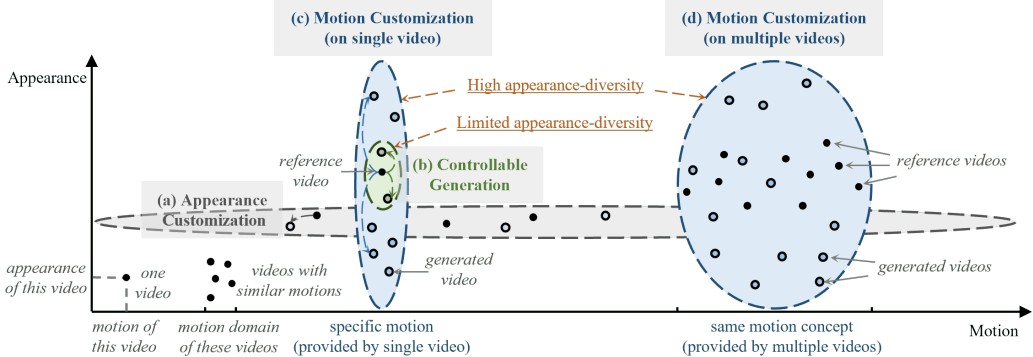

Figure 7: Each video is characterized by two aspects: appearance and motion. We can uniquely identify a video based on its values along the appearance and motion axes, as shown in the lower-left corner of this figure. (a) Appearance customization aims to create videos whose appearances look like reference videos but have different motions. (b) The controllable generation aims to generate videos with the same motion represented by control signals. However, the control singles often have constraints on appearance, limiting the appearance diversity of the generated results. (c) Motion customization on a single video aims to generate videos with the specific motion learned from reference videos while keeping the ability to generate highly diverse appearances. (d) Motion customization on multiple videos aims to learn the same motion concepts in the reference videos, such as lifting weights or playing golf, and generate videos with these motions and with highly diverse appearances.

## A.2 TRAINING AND INFERENCE DETAILS

The LoRAs are trained using Adam optimizer (Kingma & Ba, 2014), with the default betas set to $0.9$ and $0.999$, and the epsilon set to the default $1e$-$8$. We set the learning rate to $1e$-$4$, and set the weight decay to $5e$-$4$. In addition, we incorporate a dropout rate of $0.1$ during the training of LoRAs. The LoRA rank is set to $32$. For the temporal training objective, the coefficients of original temporal loss and appearance-debiased loss are both set to $1$. We sample $16$ frames with $384 \times 384$ spatial resolution and $8$ fps from each training data. The number of training steps is set to $400$ for motion customization on a single video and set to $1000$ for motion customization on multiple videos. To conserve VRAM, we employ mixed precision training with fp16. On an NVIDIA A5000 graphics card, it takes around $8$ minutes and $20$ minutes for MotionDirector to converge on a single reference video and multiple videos, respectively. On the same device, it takes $10$ minutes for Tune-A-Video (Wu et al., 2022) to converge.

In the inference stage, the diffusion steps are set to 30 and the classifier-free guidance scale is set to 12. We use the DDPM Scheduler (Ho et al., 2020) as the time scheduler in the training and inference stages.

## A.3 DETAILS OF BENCHMARKS

For the motion customization on multiple videos, we adopt the UCF Sports Action data set (Soomro & Zamir, 2015) to conduct experiments. The original data set consists of 150 videos of 10 different human motions. We remove some inconsistent videos and low-resolution ones and split one motion concept into three motion concepts, i.e. "playing golf, side view", "playing golf, front view", and "playing golf, back view", to make them more clear and more precise. Then we get in total of 95 reference videos of 12 different types of motions, where each of them has 4 to 20 reference videos. We label each type of motion with a text description, like "A person is riding a horse", and provide 6 text prompts with significant appearance changes for the generation, like "A monkey is riding a horse". An example is shown in Fig. 8.

For the motion customization on a single video, we adopt the open-sourced benchmark released by the LOVEU-TGVE competition at CVPR 2023 (Wu et al., 2023). The dataset includes 76 videos, each originally associated with 4 editing text prompts. The editing prompts focus on object change, background change, style change, and multiple changes. To further examine the diversity of text-guided generation, we add 3 more prompts that have large appearance changes for each video, such as changing man to bear or changing the outdoors to the moon. An example is shown in Fig. 8.

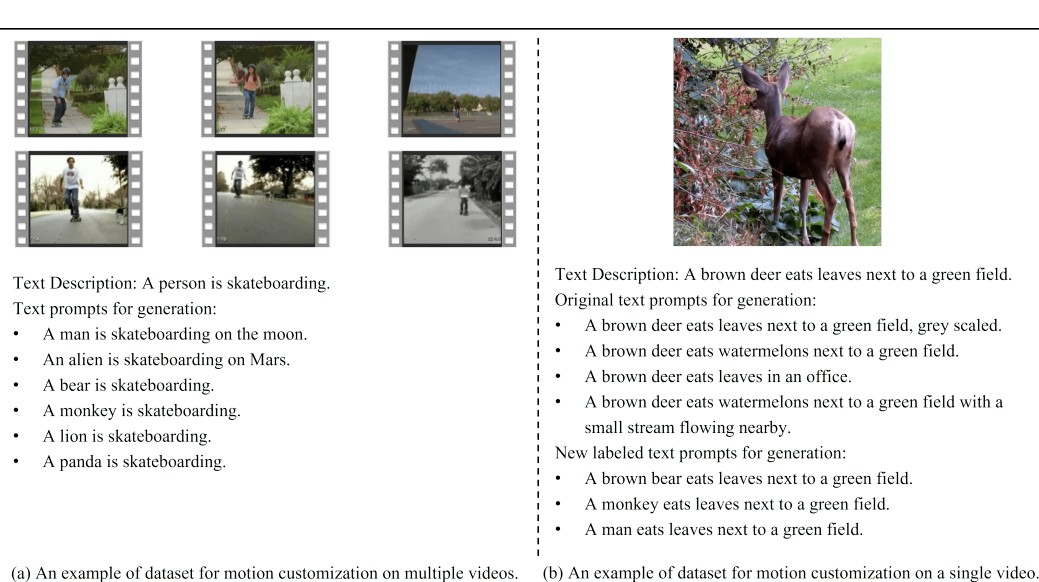

Text Description: A person is skateboarding.

Text prompts for generation:
- A man is skateboarding on the moon.
- An alien is skateboarding on Mars.
- A bear is skateboarding.
- A monkey is skateboarding.
- A lion is skateboarding.
- A panda is skateboarding.

Text Description: A brown deer eats leaves next to a green field.

Original text prompts for generation:
- A brown deer eats leaves next to a green field, grey scaled.
- A brown deer eats watermelons next to a green field.
- A brown deer eats leaves in an office.
- A brown deer eats watermelons next to a green field with a small stream flowing nearby.

New labeled text prompts for generation:
- A brown bear eats leaves next to a green field.
- A monkey eats leaves next to a green field.
- A man eats leaves next to a green field.

(a) An example of dataset for motion customization on multiple videos. (b) An example of dataset for motion customization on a single video.

Figure 8: Examples of two benchmarks for testing motion customization on multiple videos and a single video, respectively.

## A.4 DETAILS OF HUMAN EVALUATIONS

Through the Amazon MTurk platform [2], we release over 1800 comparison tasks and each task is completed by 5 human raters. Each task involves a comparison between two generated videos produced by two anonymous methods. These comparisons assess aspects such as text alignment, temporal consistency, and motion fidelity, where the motion fidelity has a reference video for raters to compare with. As shown in Fig. 9, each task has three questions for raters to answer: (1) Text alignment: Which video better matches the caption "{text prompt}"? (2) Temporal Consistency: Which video is smoother and has less flicker? (3) Motion Fidelity: Which video's motion is more similar to the motion of the reference video? Do not focus on their appearance or style, compare their motion.

---

[2] https://requester.mturk.com/

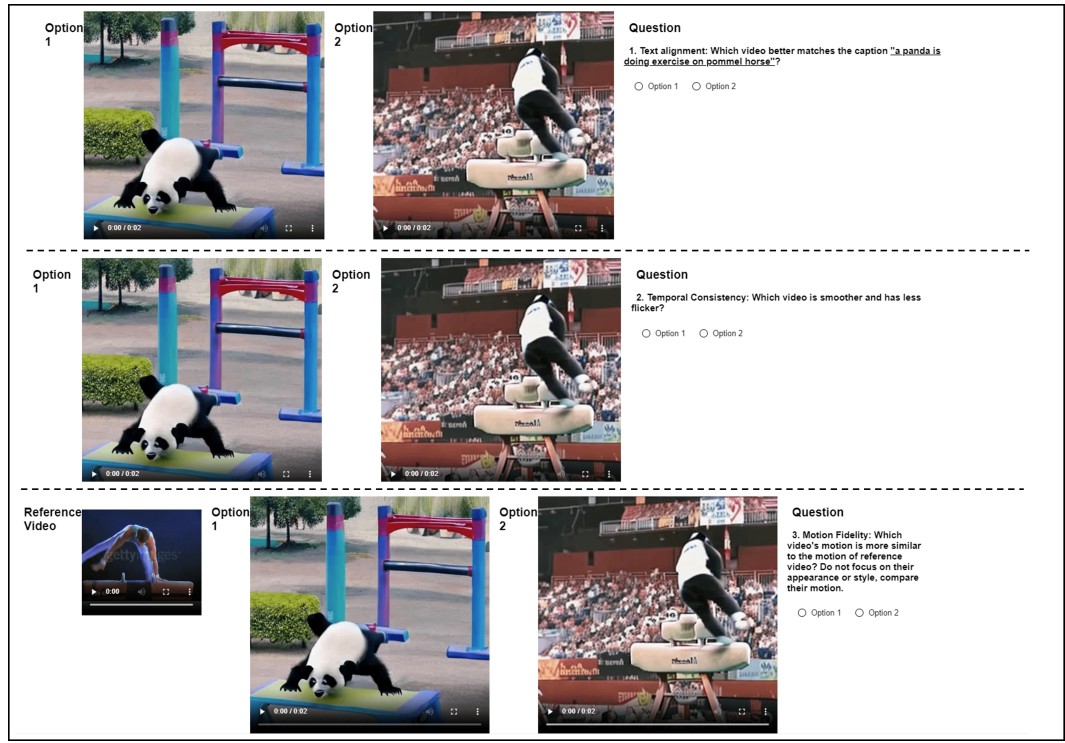

Figure 9: Example of one task for 5 human raters on Amazon MTurk to complete. Each task involves three questions comparing two generated results in terms of text alignment, temporal consistency, and motion fidelity.

For the comparison of motion customization on multiple reference videos, we use "appearance diversity" to refer to "text alignment" in Table. 1, since we use the text prompts with large appearance changes to test the methods, like changing a human to a panda. For the comparison of motion customization on a single video, the text alignment with the original text prompts of the benchmark LOVEU-TGV (Wu et al., 2023) is termed as "'text alignment" in Table. 2, while the text alignment with newly labeled text prompts with significant changes is termed as "appearance diversity". To eliminate noise in the rated results, we employ a consensus voting approach to statistically analyze the results, considering a result valid only when at least 3 out of 5 raters reach the same choice."

In the end, the released over 1800 tasks received answers from 421 unique human raters. The statistical results are listed in Table. 1 and Table. 2.

## A.5 ADDITIONAL RESULTS

We provide additional results for motion customization on multiple videos, as shown in Fig 10, and motion customization on a single video, as shown in Fig 11. Two more examples of comparison of the proposed MotionDirector and other controllable generation methods and the tuning-based method are shown in Fig 12.

Reference Videos: A person is skateboarding.

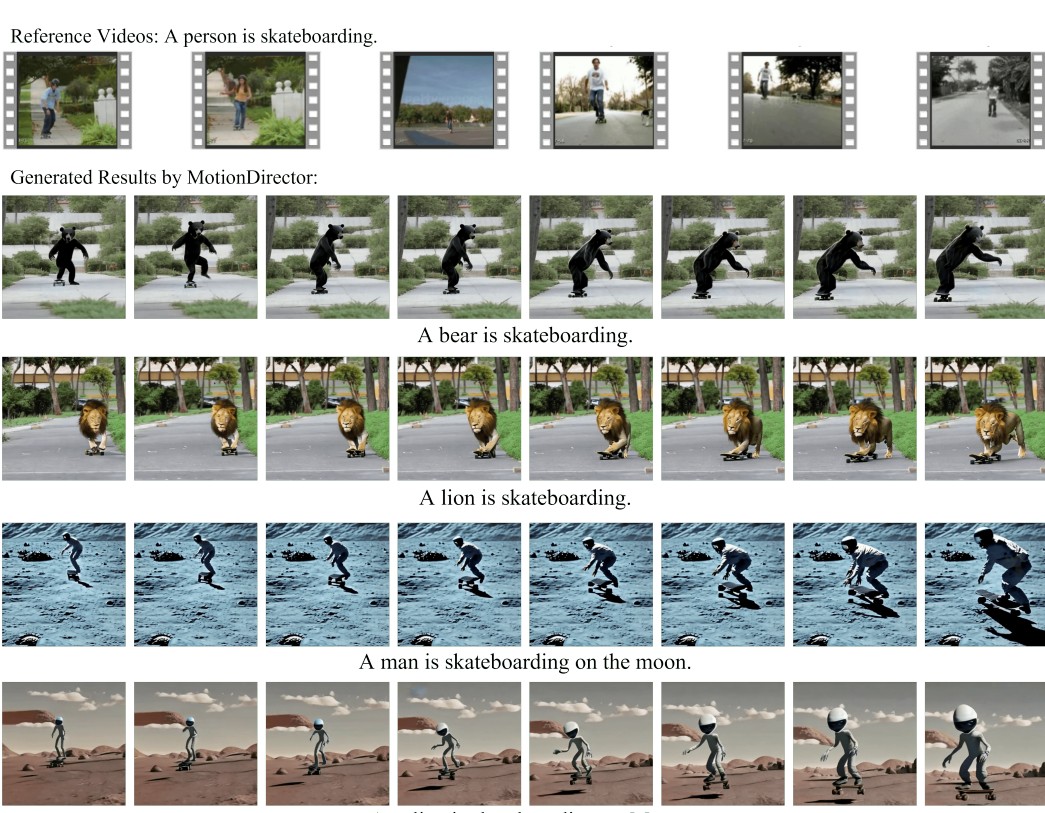

Generated Results by MotionDirector:

A bear is skateboarding.

A lion is skateboarding.

A man is skateboarding on the moon.

An alien is skateboarding on Mars.

Figure 10: Results of motion customization of the proposed MotionDirector on multiple reference videos.

Reference Videos: Surround shot of a villa in a garden.

Generated Results by MotionDirector:

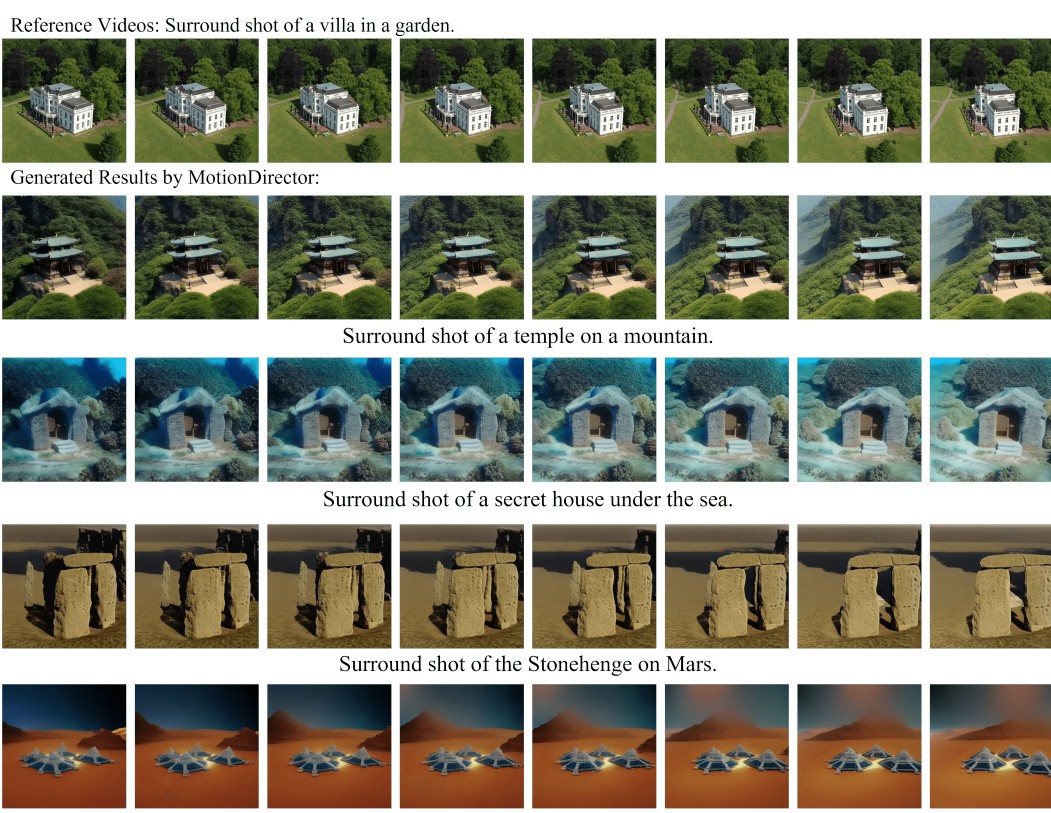

Surround shot of a temple on a mountain.

Surround shot of a secret house under the sea.

Surround shot of the Stonehenge on Mars.

Surround shot of an alien base on Mars.

Figure 11: Results of motion customization of the proposed MotionDirector on a single reference video.

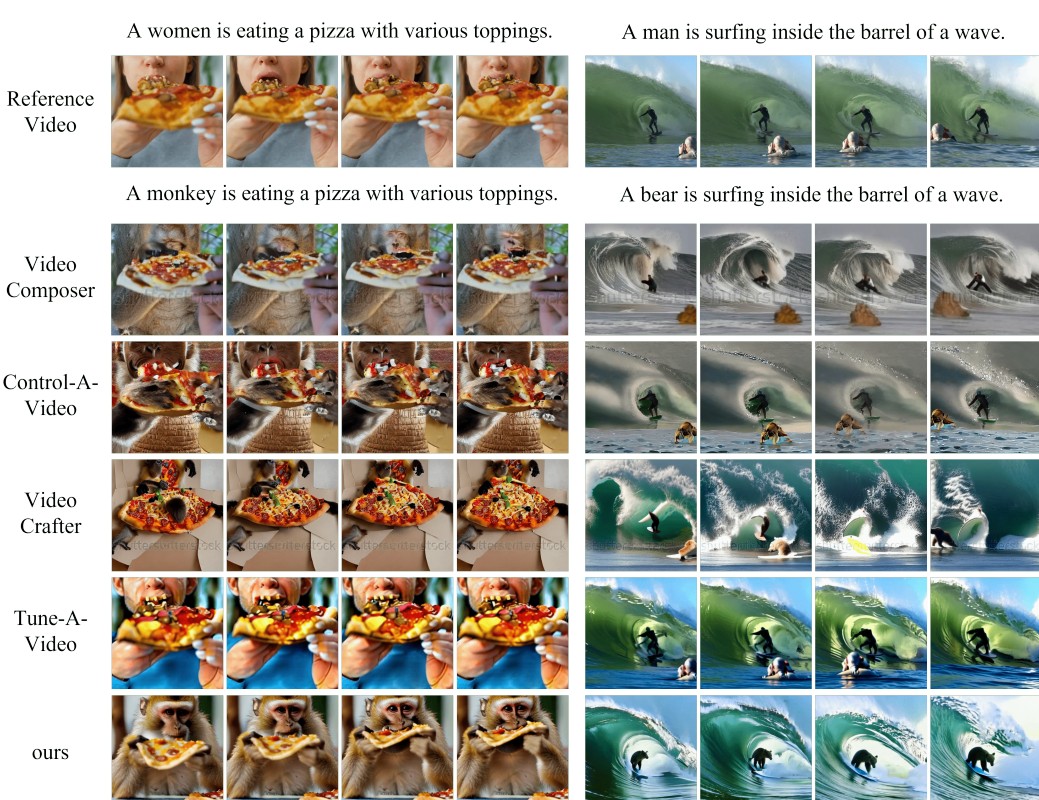

Figure 12: Comparison of different methods on the task of motion customization given a single reference video.

