# OpenReview forum: "MotionDirector: Motion Customization of Text-to-Video Diffusion Models"
_ICLR.cc/2024/Conference — ICLR 2024 Conference Withdrawn Submission_

### Official Review · Reviewer_LtG6 · 2023-10-26

**Soundness:** 3 good
**Presentation:** 3 good
**Contribution:** 3 good
**Rating:** 6
**Confidence:** 4

**Summary:**

The task of Motion Customization involves adapting these models to produce videos showcasing specific motions using reference video clips. However, conventional adaptation methods often entangle motion concepts with appearances, limiting customization. To address this, "MotionDirector" is introduced, employing a dual-path Low-Rank Adaptions (LoRAs) architecture and an appearance-debiased temporal loss, which effectively decouples appearance and motion, enabling more versatile video generation.

**Strengths:**

- The paper identifies the challenge in generalizing customized motions across diverse appearances. The integration of motion in the video appears great, and this effect can be attributed to the decoupling treatment of the temporal module.
- It proposes a dual-path architecture designed to separate the learning of appearance and motion.
- The visual results show that the proposed method outperforms multiple baseline methods on motion control and video object replacement.

**Weaknesses:**

- The explanation of the decentralized temporal loss is not very clear. It might be beneficial to verify the effect of this loss through more ablation experiments, especially in the context of video visualization.
- Training LoRA does not appear to be computationally intensive, but it's advisable to specify the training cost in the article.
- There are concerns about the generalizability of this method for video motion extraction. It's worth considering the possibility of developing a unified video motion extraction module to address this issue.
- There are no supplementary videos, which makes the paper less convincing.

**Questions:**

See above

**Details Of Ethics Concerns:**

The generated content may contain some concept bias.

---

> ### Author Response · Authors · 2023-11-15
> **Response to Reviewer LtG6 (1/3)**
>
> **Q1: About the explanation of the proposed loss.**
> >Weakness 1: The explanation of the decentralized temporal loss is not very clear. It might be beneficial to verify the effect of this loss through more ablation experiments, especially in the context of video visualization.
>
> **A1:** Thanks for your suggestion. We will add more detailed explanations, discussions, and experimental analysis as follows.
>
> Explanation:
>
> Besides Fig. 4, Equation 5, and Equation 6, we will add more explanations of the motivations and the design of the appearance-debiased temporal loss. In Fig. 4, we take 4 videos as an example, by visualizing their latent codes, we find that the internal connectivity structure between latent codes is more influenced by motion, while the distance between sets of latent codes is primarily affected by the difference in appearance. It is easy to notice that the latent codes of videos with similar motions but different appearances lie in quite different locations. This kind of bias caused by appearance differences makes it difficult for models to learn motions by directly training in the original latent space. To eliminate the appearance bias and further ease the learning of the motions we propose the appearance-debiased operation as formulated in Equation 5. The frames from the same video have similar appearances while the motion dynamics among them slightly change their latent codes, so their latent codes lie in relatively close locations. On the other hand, since these frames have similar appearances, electing one frame as an anchor and subtracting it from all frames can mitigate the influence of appearance on the learning of motions, to a large extent. Thus we have the form of Equation 5.
>
> Experimental Analysis:
>
> To test the effectiveness of the proposed appearance-debiased temporal loss, we have conducted ablation studies and the results are shown in section 4.1. The visualization in Fig. 5 shows that compared to the method without the appearance-debiased loss (tuned w/o AD-Loss), the method with the appearance-debiased loss (ours) learns the motion concepts better. We further test the methods on 95 videos of 12 different human motions, and the results are listed in Table 1. As shown in the last column ‘Motion Fidelity’ of the Table. 1, the method "w/o AD-Loss" only gains a slight improvement compared with the foundation model, while the method with appearance-debiased loss (ours) achieves improvements with larger gaps. This justifies the effectiveness of the proposed appearance-debiased temporal loss.

---

> ### Author Response · Authors · 2023-11-15
> **Response to Reviewer LtG6 (2/3)**
>
> **Q2: About the training cost.**
> >Weakness 2: Training LoRA does not appear to be computationally intensive, but it's advisable to specify the training cost in the article.
>
> **A2:** Thanks for your advisation. We have provided the discussion of training costs in section 4.3 as follows.
>
> The lightweight LoRAs enable our method to tune the foundation models efficiently. Taking the foundation model ZeroScope for example, it has over 1.8 billion pre-trained parameters. Each set of trainable spatial and temporal LoRAs only adds 9 million and 12 million parameters, respectively. Requiring 14 GB VRAM, MotionDirector takes 20 minutes to converge on multiple reference videos, and 8 minutes for a single reference video, competitive to the 10 minutes required by Tuna-A-Video [1].
>
> [1] Wu, Jay Zhangjie, et al. "Tune-a-video: One-shot tuning of image diffusion models for text-to-video generation." Proceedings of the IEEE/CVF International Conference on Computer Vision. 2023.

---

> ### Author Response · Authors · 2023-11-15
> **Response to Reviewer LtG6 (3/3)**
>
> **Q3: About the motion extraction module.**
> >Weakness 3: There are concerns about the generalizability of this method for video motion extraction. It's worth considering the possibility of developing a unified video motion extraction module to address this issue.
>
> **A3:** Thanks for your valuable suggestions. Building a unified motion extraction module is a good idea. We have tried train models on the optical flows extracted by the pre-trained motion extraction model. However, the results are influenced heavily by the quality of motion extraction, making it hard to generalize to some complex videos. Besides, it requires more computing resources to adapt foundational models to take in additional information such as optical flows. In contrast, our models are specifically tuned on a set of videos with similar motion concepts, facilitating better generalization.
>
> Thanks again for this valuable suggestion. In the future, we will continue to try to build such a powerful unified motion extraction module.
>
> **Q4: About supplementary videos.**
> >Weakness 4: There are no supplementary videos, which makes the paper less convincing.
>
> **A4:** Thanks for pointing this out. The supplementary videos are now provided on the anonymous page: https://motiondirector2023.github.io/. We will add this link in the revised paper.

---

> ### Author Response · Authors · 2023-11-16
> **Follow-up on Initial Rebuttal Submission**
>
> Dear Reviewer LtG6,
>
> Thank you for your valuable feedback on our submission. We have read your comments carefully and have addressed them in our rebuttal. We would be grateful if you could acknowledge if our responses have addressed your comments. We would also be happy to engage in further discussions if needed. Thank you again for your time and consideration.
>
> Best regards,
>
> Authors of paper 3534.

---

### Official Review · Reviewer_8uoP · 2023-10-28

**Soundness:** 2 fair
**Presentation:** 2 fair
**Contribution:** 2 fair
**Rating:** 3
**Confidence:** 4

**Summary:**

The paper introduces the concept of Motion Customization in text-to-video diffusion models and proposes a method called MotionDirector, which applies dual-path inserted LoRAs to decouple the learning of content and motion. It also incorporates an appearance-debiased temporal loss to refine the learning process further. The paper validates the approach through experiments on two benchmarks, demonstrating its superiority in terms of motion fidelity and appearance diversity.

**Strengths:**

- The paper is well-written and easy to follow.
- The proposed appearance-debiased temporal loss sounds reasonable.
- I appreciate Fig 4, which shows the denoising paths of different conditions.
- The demo quality is good.

**Weaknesses:**

1. **Limited Novelty**:
   - The concept of decoupling the learning of content and motion is not new and has been explored in works like "Align your Latents" [1] by NVIDIA. The dual-path architecture seems to be a reiteration of this idea.
   - The methodology largely builds upon existing techniques like Low-Rank Adaptions (LoRAs).

2. **Lack of Justification for Appearance-Debiased Temporal Loss**:
   - The paper introduces an appearance-debiased temporal loss but does not provide a thorough explanation or justification for its effectiveness.
   - The introduction of a hyperparameter $\beta$ is not accompanied by a sensitivity analysis, leaving its impact on the model's performance unclear.

3. The video length is too small (number of frames), it seems only experiments on video length equal to 16 are conducted. Considering that it requires 8 minutes to fit 16 frames, it becomes very time-consuming and even inapplicable for longer videos.

4. Technical contribution is weak.

[1] Align your Latents: High-Resolution Video Synthesis with Latent Diffusion Models

**Questions:**

See Weaknesses.

---

> ### Author Response · Authors · 2023-11-14
> **Response to Reviewer 8uoP (1/4)**
>
> **Q1: About the novelty.**
>
> **Q1.1: Differences with "Align your Latents".**
> >Weakness 1.1: The concept of decoupling the learning of content and motion is not new and has been explored in works like "Align your Latents" [1] by NVIDIA. The dual-path architecture seems to be a reiteration of this idea.
>
> **A1.1:** The work "Align your Latents" proposed a video latent diffusion model (Video-LDM) to achieve text-to-video generation. This work proposes to first pre-train an LDM on images and then add temporal layers to spatial layers in cascade and train temporal layers on videos.
>
> The work "Align your Latents" is different from ours in both task and method:
>
> Tasks are different: "Align your Latents" is a foundational text-to-video model that aims to learn video generation ability by pre-training on large-scale video datasets, such as the WebVid-10M consists of 10.7M video-caption pairs. The motions in its generated videos can only be controlled with text prompts. In contrast, our work, MotionDirector, aims to customize such a foundation model to generate videos with user-desired motions. The customization task is different from the pre-training foundational models, just like the DreamBooth [1], which is famous for customizing appearance in text-to-image generations, is different from the stable diffusion [2], which is a foundational text-to-image generation model. More details about the difference between the pre-training foundational models and customizing them can be found in the first two paragraphs of the introduction section.
>
> Methods are different: "Align your Latents" adds new temporal layers to latent diffusion models with only spatial layers. The temporal layers are connected after the spatial layers and are trained on millions of videos. In contrast, our work, MotionDirector, builds upon pre-trained foundational models that already incorporate both spatial and temporal layers. We fix all the pre-trained parameters to keep the learned generation ability,  inject LoRAs to spatial and temporal layers, and train them in a dual-path paradigm to learn the motions decoupled from the limited appearance provided by a small set of reference videos.  It is worth mentioning that, "Align your Latents" does not have any dual-path architecture.
>
> **Q1.2: The use of LoRAs.**
> >Weakness 1.2: The methodology largely builds upon existing techniques like Low-Rank Adaptions (LoRAs).
>
> **A1.2:** The Low-Rank Adaptions (LoRAs) have become a kind of foundational tool that could achieve efficient fine-tuning of large pre-trained models. Our work focuses on how to customize the pre-trained foundational models to generate desired motions, rather than reinvent such basic fine-tuning tools as LoRAs. Some cutting-edge publications, such as Dreambooth [1], and Mix-of-show [3],  also use these basic fine-tuning tools to achieve their new idea. To achieve the task of motion customization, our contribution lies in proposing a dual-path architecture and a novel appearance-debiased temporal training objective, to learn the motions better and avoid the influence of limited appearances provided by a small set of reference videos, to overcome the challenge of generalizing the learned motions to diverse appearances.
>
>
> [1] Ruiz, Nataniel, et al. "Dreambooth: Fine-tuning text-to-image diffusion models for subject-driven generation." Proceedings of the IEEE/CVF Conference on Computer Vision and Pattern Recognition. 2023.
>
> [2] Rombach, Robin, et al. "High-resolution image synthesis with latent diffusion models." Proceedings of the IEEE/CVF conference on computer vision and pattern recognition. 2022.
>
> [3] Gu, Yuchao, et al. "Mix-of-Show: Decentralized Low-Rank Adaptation for Multi-Concept Customization of Diffusion Models." Advances in Neural Information Processing Systems. 2023.

---

> > ### Comment · Reviewer_8uoP · 2023-11-15
> >
> > Thank you for your reply. Any updates for defending?

---

> > > ### Author Response · Authors · 2023-11-15
> > > **Response to Reviewer 8uoP**
> > >
> > > Dear Reviewer 8uoP,
> > >
> > > We appreciate your time and effort in reviewing our paper. Looking forward to your feedback, and we are very happy to discuss if you have any further questions.
> > >
> > >
> > > Best regards,
> > >
> > > Authors of paper 3534.

---

> ### Author Response · Authors · 2023-11-15
> **Response to Reviewer 8uoP (2/4)**
>
> **Q2: Justification for Appearance-Debiased Temporal Loss.**
>
> **Q2.1: Explanation or Justification of Effectiveness.**
> >Weakness 2.1: The paper introduces an appearance-debiased temporal loss but does not provide a thorough explanation or justification for its effectiveness.
>
> **A2.1:** Explanation:
>
> On page 6 and in Equation 5 and Fig. 4, we have provided the motivation, the visualization, and the formulation of the proposed appearance-debiased temporal loss. In Fig. 4, we take 4 videos as an example, by visualizing their latent codes, we find that the internal connectivity structure between latent codes is more influenced by motion, while the distance between sets of latent codes is primarily affected by the difference in appearance. It is easy to notice that the latent codes of videos with similar motions but different appearances lie in quite different locations. This kind of bias caused by appearance makes it difficult for models to learn motions by directly training in the original latent space. To eliminate the appearance bias and further ease the learning of the motions we propose the appearance-debiased operation as formulated in Equation 5. The frames from the same video have similar appearances while the motion dynamics among them slightly change their latent codes, so their latent codes lie in relatively close locations. On the other hand, since these frames have similar appearances, electing one frame as an anchor and subtracting it from all frames can mitigate the influence of appearance on the learning of motions, to a large extent. Thus we have the form of Equation 5.
>
> We will add a more detailed explanation as mentioned above to help readers better understand the proposed appearance-debiased loss.
>
> Justification:
>
> To test the effectiveness of the proposed appearance-debiased temporal loss, we have conducted ablation studies and the results are shown in section 4.1. Fig. 5 shows that compared to the method without the appearance-debiased loss (tuned w/o AD-Loss), the method with the appearance-debiased loss (ours) learns the motion concepts better. We further test the methods on 95 videos of 12 different human motions, and the results are listed in Table 1. As shown in the last column ‘Motion Fidelity’ of the Table. 1, the method "w/o AD-Loss" only gains a slight improvement compared with the foundation model, while the method with appearance-debiased loss (ours) achieves improvements with larger gaps. This justifies the effectiveness of the proposed appearance-debiased temporal loss.
>
> **Q2.2: About the hyperparameter.**
> > Weakness 2.2: The introduction of a hyperparameter $\beta$ is not accompanied by a sensitivity analysis, leaving its impact on the model's performance unclear.
>
> **A2.2:** In Equation 5, since $\epsilon_i$ and $\epsilon_{anchor}$ follow Gaussian distributions, the $\phi(\epsilon_i)$ also follow Gaussian distribution. Thus the appearance-debiased operation does not change the distribution of the latent codes, which ensures the stability of the training process and is not sensitive to the value of $\beta$. On the other hand, the value of $\beta$ determines how much the anchor is subtracted from each frame and how easily the motions can be learned. We consider appearance and motion as the two main components of the video and assume that their importance is equal. Thus we have $\epsilon_i = \frac{1}{\sqrt{2}} \epsilon_{appearance}+  \frac{1}{\sqrt{2}} \epsilon_{motion}$. Since the appearance can be represented by the anchor to a large extent, it is natural to have $\epsilon_i  = \frac{1}{\sqrt{2}} \epsilon_{anchor} + \frac{1}{\sqrt{2}} \phi(\epsilon_i)$, which is same to $\phi(\epsilon_i) = \sqrt{2} \epsilon_i  - \epsilon_{anchor} $, which is setting $\beta$ equal to 1 in Equation 5. The experimental results show that setting the value of $\beta$ to 1 can achieve better performance compared with baseline methods.

---

> ### Author Response · Authors · 2023-11-15
> **Response to Reviewer 8uoP (3/4)**
>
> **Q3: About the video length.**
> >Weakness 3: The video length is too small (number of frames), it seems only experiments on video length equal to 16 are conducted. Considering that it requires 8 minutes to fit 16 frames, it becomes very time-consuming and even inapplicable for longer videos.
>
> **A3:** Long video generation is another raising research area that is still under-explored. In this paper, we focus on motion customization of foundation models rather than retraining a more powerful foundation model that can generate longer videos. Current open-sourced foundational text-to-video diffusion models target generating videos with 8 to 32 frames, and their temporal layers are trained on videos of such length. Since our method is built upon them, our generated videos are also of such length. If the pre-trained temporal interpolation model is applied, just like some foundational models do, the number of frames can be further extended.
>
> For the tuning speed, our method has a similar performance to the tuning-based method, such as the Tune-A-Video [4]. Given that current customization tasks often need tuning the foundational models to achieve good performance, and LoRAs are one of the most efficient methods among the tuning methods, we believe the tuning speed of MotionDirector is acceptable.
>
> [4] Wu, Jay Zhangjie, et al. "Tune-a-video: One-shot tuning of image diffusion models for text-to-video generation." Proceedings of the IEEE/CVF International Conference on Computer Vision. 2023.

---

> ### Author Response · Authors · 2023-11-15
> **Response to Reviewer 8uoP (4/4)**
>
> **Q4: About the technical contribution.**
> >Weaknesse 4: The video length is too small (number of frames), it seems only experiments on video length equal to 16 are conducted. Considering that it requires 8 minutes to fit 16 frames, it becomes very time-consuming and even inapplicable for longer videos.
>
> **A4:**  Our contributions are in two aspects.
>
> First, to the best of our knowledge, we are the first to propose and define the task of motion customization in video generation. Given a set of video clips of the same motion concept, the task of Motion Customization is to adapt existing text-to-video diffusion models to generate videos with this motion. We point out that the challenge lies in generalizing the customized motions to various appearances.
>
> Second, to overcome this challenge, we propose the MotionDirector with a dual-path architecture and a novel appearance-debiased temporal training objective, to decouple the learning of appearance and motion. The dual-path LoRAs and the appearance-debiased temporal training objective are novel and helpful for learning motions decoupled from limited appearances in a small set of reference videos. The experimental results demonstrate their effectiveness.

---

> ### Comment · Reviewer_8uoP · 2023-11-15
>
> Hi, thanks for the explanation. You treat $ \boldsymbol{\epsilon_{apperance}}$  as  $ \boldsymbol{\epsilon_{anchor}}$, right?

---

> > ### Comment · Reviewer_8uoP · 2023-11-15
> >
> > To be honest and with no offense, I don't think the explanation for Appearance-Debiased Temporal Loss is reasonable. If my understanding is correct, you treat $\boldsymbol{\epsilon_{appearnce}}$ as $\boldsymbol{\epsilon_{anchor}}$, which assumes that when adding losses for training, the added noises in different frames are correlated. However, the noises for different frames are independently sampled following normal distribution.

---

> > > ### Author Response · Authors · 2023-11-15
> > > **Response to Reviewer 8uoP**
> > >
> > > Thanks for your feedback. In the training stage, the noises added to different frames are still noises independently sampled following normal distribution. In other words, the input format and output format of the model remain unchanged. The appearance-debiased operation is only applied when we calculate the temporal loss as Equation 6. This can eliminate the influence of appearances on the temporal loss calculation to a large extent and will not harm the stability of the training process.

---

> > > > ### Comment · Reviewer_8uoP · 2023-11-15
> > > >
> > > > Yeah. And this is exactly what I am concerned about. Your explanation doesn't align with your implementation.

---

> > > > > ### Author Response · Authors · 2023-11-15
> > > > > **Response to Reviewer 8uoP**
> > > > >
> > > > > Could you please specify which aspect of the implementation you are referring to? Is the part on page 6 before equation 5? Our intention in presenting this part is to provide Equation 5 to help understand the $\phi$ in Equation 6. We will add a clearer explanation of ''the input noise will not be changed'' in the revised paper.

---

> > > > > > ### Comment · Reviewer_8uoP · 2023-11-15
> > > > > >
> > > > > > For the explanation:
> > > > > >
> > > > > > According to your explanation, you assume that the noises added to different frames are correlated. Because you said
> > > > > >
> > > > > > > $\epsilon_{i}$ can be written as the weighted addition of $\epsilon_{motion}$ and $\epsilon_{apperance}$.
> > > > > >
> > > > > > Then, you simply replace $\epsilon_{motion}$ with $\epsilon_{anchor}$ which is the noise added to one frame, right?
> > > > > >
> > > > > > So, in practice, you have assumed that the noise added to different frames is correlated.
> > > > > >
> > > > > > ---
> > > > > >
> > > > > > For the implementation:
> > > > > > You said
> > > > > >
> > > > > > > In the training stage, the noises added to different frames are still noises independently sampled following normal distribution.
> > > > > >
> > > > > > ---
> > > > > >
> > > > > > Hence, I think they are not aligned if my understanding is correct.

---

> ### Author Response · Authors · 2023-11-15
> **Response to Reviewer 8uoP**
>
> Thanks for your feedback. Please notice that the noises added to frames as input are not changed, and are not correlated. The appearance-debiasing is only applied to noises when we calculate the temporal loss after the model predicts the noises.
>
> To be more specific, the noises added to frames are independently sampled following normal distribution, without any appearance-debiasing operation. Then, the model takes in the noisy latent codes and predicts the estimated noises. When we calculate the appearance-debiased temporal loss based on the difference between the input noises and the estimated noises, we apply appearance-debiasing on both of them. Through such "post-processing", most of the loss comes from the errors in motion prediction rather than the appearance prediction.

---

> ### Author Response · Authors · 2023-11-15
> **Response to Reviewer 8uoP**
>
> Thanks for your questions and the in-depth discussion above.
>
> We add more details to the A2.2 as follows to provide a clearer explanation. We hope this can address your concerns.
>
> Explanation:
>
> We consider appearance and motion as the two main components of the video. Thus we assume:
>
> $z_{t,i} = \frac{\beta}{\sqrt{\beta^2+1}} z_{t,i}^{appearance}+  \frac{1}{\sqrt{\beta^2+1}} z_{t,i}^{motion}$, (a)
>
> where $z_{t,i}$ is the latent code of frame $i$ at time step $t$.
>
> Since the appearance can be represented by the anchor to a large extent, we have:
>
> $z_{t,i}  = \frac{\beta}{\sqrt{\beta^2+1}}z_{t, anchor} + \frac{1}{\sqrt{\beta^2+1}} z_{t,i}^{motion}$,
>
> which is the same to:
>
>  $z_{t,i}^{motion} = \phi(z_{t,i}) = \sqrt{\beta^2+1} z_{t,i} - \beta z_{t,anchor}$. (b)
>
> Since we have $z_t = \sqrt{\bar{\alpha_t}}z_0 + \sqrt{1-\bar{\alpha_t}}\epsilon$ as in Equation 2, where $\epsilon$ represents the noises at time step $t$ and $\epsilon\sim \mathcal{N}(0, 1)$. The Equation (b) can be written as:
>
>  $\sqrt{\bar{\alpha_t}}z_{t=0,i}^{motion} + \sqrt{1-\bar{\alpha_t}}\epsilon_i^{motion}  = \sqrt{\beta^2+1} (\sqrt{\bar{\alpha_t}}z_{t=0,i} + \sqrt{1-\bar{\alpha_t}}\epsilon_i) - \beta (\sqrt{\bar{\alpha_t}}z_{t=0,anchor} + \sqrt{1-\bar{\alpha_t}}\epsilon_{anchor})$,
>
> =>
>
>  $\sqrt{\bar{\alpha_t}}z_{t=0,i}^{motion} + \sqrt{1-\bar{\alpha_t}}\epsilon_i^{motion}  = \sqrt{\bar{\alpha_t}} ( \sqrt{\beta^2+1} z_{t=0,i} - \beta z_{t=0,anchor}) + \sqrt{1-\bar{\alpha_t}} (\sqrt{\beta^2+1} \epsilon_{i} - \beta \epsilon_{anchor}). $ (c)
>
> Since $z_{t=0}$ is a constant, we can rewrite the equation (c) as:
>
> $C_1 + \epsilon_i^{motion} = C_2 + \sqrt{\beta^2+1} \epsilon_{i} - \beta \epsilon_{anchor}$
>
> =>
>
> $\epsilon_i^{motion} = \sqrt{\beta^2+1} \epsilon_{i} - \beta \epsilon_{anchor} + C_3,$ (d)
>
> where $C_1$, $C_2$, and $C_3$ are constants that do not depend on the parameters $\theta$ of the model. So, when we calculate loss to optimize parameters, the equation (d) can be simplified as follows,
>
> $\epsilon_i^{motion} = \phi(\epsilon_i) = \sqrt{\beta^2+1} \epsilon_{i} - \beta \epsilon_{anchor},$ (e)
>
> which is exactly the Equation 5 in the paper.
>
> In the above process, all assumptions are about latent codes. The noises $\epsilon$ are independently sampled following the normal distribution $\epsilon\sim \mathcal{N}(0, 1)$. For the value of $\beta$ in equation (a), if we naturally assume that the importance of appearance and motion is equal, then we get $\beta=1$.
>
> Implementation:
>
> * Input: The noises added to frames are independently sampled following the normal distribution: $\epsilon\sim \mathcal{N}(0, 1)$.
>
> * Output: The estimated noises predicted by the model: $\epsilon_\theta(z_t, t, \tau_\theta(y))$.
>
> * Loss calculation: When we calculate loss, we apply the proposed appearance-debiasing to eliminate the influence of appearances. Following the above explanation, we have the equations (a) to (e), and formulate the appearance-debiasing as $\phi(\cdot)$. Thus the loss is calculated as
> $ \mathcal{L} = \mathbb{E} \left[ \lVert \phi(\epsilon) - \phi(\epsilon_\theta(z_t, t, \tau_\theta(y))) \rVert_2^2 \right] $,
> which is Equation 6 in the paper.
>
> In conclusion, we believe the above explanation and implementation are aligned with each other.

---

> ### Author Response · Authors · 2023-11-16
>
> Dear Reviewer 8uoP,
>
> We would like to express our gratitude for the effort and time you have dedicated to the review process. We hope that all your concerns have been addressed. If you have any additional questions, we would be delighted to discuss them with you.
>
> Best regards,
>
> Authors of paper 3534.

---

> ### Author Response · Authors · 2023-11-17
> **Response to Reviewer 8uoP**
>
> Dear Reviewer 8uoP,
>
> We sincerely appreciate the time and effort you have invested in reviewing our paper.
>
> We would like to inquire whether our response has addressed your concerns and if you have the time to provide further feedback on our rebuttal. We are more than willing to engage in further discussion.
>
> Best regards,
>
> Authors of paper 3534.

---

> > ### Comment · Reviewer_8uoP · 2023-11-17
> >
> > Busy right now, I will respond to you later. Thank you for your efforts.

---

> > > ### Author Response · Authors · 2023-11-17
> > > **Response to Reviewer 8uoP**
> > >
> > > Dear Reviewer 8uoP,
> > >
> > > Thank you for your response and the time you dedicated to reviewing. We are genuinely looking forward to further discussion with you.
> > >
> > > Best regards,
> > >
> > > Authors of paper 3534.

---

> > > ### Author Response · Authors · 2023-11-18
> > > **Response to Reviewer 8uoP**
> > >
> > > Dear Reviewer 8uoP,
> > >
> > > Thank you for your efforts in the review. Could you please let us know if your concerns regarding points 1, 3, and 4 have been addressed? Thank you very much.
> > >
> > > Best regards,
> > >
> > > Authors of paper 3534.

---

### Official Review · Reviewer_QKrd · 2023-10-28

**Soundness:** 3 good
**Presentation:** 3 good
**Contribution:** 3 good
**Rating:** 8
**Confidence:** 2

**Summary:**

This paper presented MotionDirector, a diffusion-based pipeline for text-to-video video editing. The paper emphasized the design of motion-appearance decoupling dual-path architecture and a special appearance-de-biased temporal loss. The experiments are conducted on UCF Sports and LOVEU-TGVE-2023.

**Strengths:**

1. The way that authors decouple the motion and appearance when using LoRA is novel and smart.
2. The qualitative experiment results are convincing.
3. The paper generally reads well.

**Weaknesses:**

1. I personally do not see a necessity that especially formulates the task of motion customization. It is a subset of video editing tasks. Meanwhile, the motion pattern is not generated from scratch nor adjustable.
2. There is no discussion of failure cases, which can provide important insights for the video editing field.

**Questions:**

1. I really love the motivation of appearance-debiased temporal loss. Especially, the illustration in Figure 4 is intriguing and meaningful. However, I expect the authors to provide more discussion and analysis for this part. Including but not limited to answering the following questions:

	* Is there a more theoretical and/or experimental proof for the hypothesis: motion primarily impacts the underlying dependencies between these point sets, whereas the distances between different sets of points are more influenced by appearance?

	* Is there a better way to evaluate the effectiveness of AD loss? In the paper, there are only two sample videos showcasing the impact of adding AD loss to the training.

2. Do the authors test how many frames can be consistently generated using the proposed method?

---

> ### Author Response · Authors · 2023-11-14
> **Response to Reviewer QKrd (1/4)**
>
> Thank you very much for recognizing our work and the valuable comments! We have prepared a comprehensive response and hope it satisfactorily addresses your concerns.
>
> **Q1: Differences between tasks.**
> >Weaknesses1: I personally do not see a necessity that especially formulates the task of motion customization. It is a subset of video editing tasks. Meanwhile, the motion pattern is not generated from scratch nor adjustable.
>
> **A1:** The task of video editing focuses on editing a single video at one time. The task of motion customization focuses on learning a motion concept which is provided by a set of reference videos, and then generalizing this motion concept to diverse appearances.
>
> These two tasks are different from each other in three aspects.
>
> 1) Their targets are different: Video-to-video editing aims to change subjects in videos to different subjects or change the style of videos. Motion customization aims to generate learned motions with diverse appearances to generate videos. Just like the DreamBooth [1], an appearance customization method, is different from the image editing methods, like the image-to-image mode of stable diffusion. DreamBooth aims to generate the learned specific subject in diverse images, while image-to-image editing aims to edit the subjects to other different subjects or change the styles of images.
>
> 2) Their input and output are different: Video editing is a single-video-to-single-video task. Motion customization is a multiple-videos-to-multiple-videos task. Motion customization requires learning motion concepts represented by a set of videos. Once the motion customization model is trained, it can be applied to generate videos with diverse appearances and the learned specific motion. Taking a single video as a reference is just a special case of motion customization.
>
> 3) Their challenges are different: Video editing requires keeping all the motion dynamics unchanged and editing the appearances of all parts or some parts of the input video. The balance of temporal consistency, flexibility of editing, and appearance authority is challenging. Motion customization requires generalizing the learned motions to diverse appearances. Learning the motions better from a small set of videos with limited appearance is challenging.
>
> [1] Ruiz, Nataniel, et al. "Dreambooth: Fine-tuning text-to-image diffusion models for subject-driven generation." Proceedings of the IEEE/CVF Conference on Computer Vision and Pattern Recognition. 2023.
>
> [2] Rombach, Robin, et al. "High-resolution image synthesis with latent diffusion models." Proceedings of the IEEE/CVF conference on computer vision and pattern recognition. 2022.

---

> ### Author Response · Authors · 2023-11-14
> **Response to Reviewer QKrd (2/4)**
>
> **Q2: About failure cases**
> > Weakness 2: There is no discussion of failure cases, which can provide important insights for the video editing field.
>
> **A2:** Most failure cases appear when the MotionDirector is applied to learn the complicated motions of multiple objects in the videos, such as a group of boys playing soccer. We provided some discussion of this case in section 5. We think a possible solution is to further decouple the motions of different subjects in the latent space and learn them separately.

---

> ### Author Response · Authors · 2023-11-14
> **Response to Reviewer QKrd (3/4)**
>
> **Q3: More discussion and analysis for appearance-debiased temporal loss.**
> >Question1: I really love the motivation of appearance-debiased temporal loss. Especially, the illustration in Figure 4 is intriguing and meaningful. However, I expect the authors to provide more discussion and analysis for this part. Including but not limited to answering the following questions:
>
> **A3:** Thanks for your recognition and the valuable suggestion. We will add more discussion and analysis for the appearance-debiased temporal loss as follows.
>
> **Q3.1: About the hypothesis.**
> >Question 1.1: Is there a more theoretical and/or experimental proof for the hypothesis: motion primarily impacts the underlying dependencies between these point sets, whereas the distances between different sets of points are more influenced by appearance?
>
> **A3.1:** Compared with those frames from different videos, the frames from the same video have much more similar appearances while the motion dynamics among them slightly change their latent codes, so their latent codes lie in relatively close locations. The appearances of frames from different videos are more different from each other, thus the distances between different sets of points are much larger than the inner distances of each set. Thus we claim that the distance between clusters is primarily affected by the difference in appearance, while the motion impacts the internal connectivity structure, the underlying dependencies, in these point sets.
>
> **Q3.2: About the evaluation of the effectiveness of AD loss.**
> >Question 1.2: Is there a better way to evaluate the effectiveness of AD loss? In the paper, there are only two sample videos showcasing the impact of adding AD loss to the training.
>
> **A3.2:** Yes. Besides the two examples in Fig. 5, we conducted comparison experiments on 95 videos of 12 different human motions to test the effectiveness of AD-Loss. The evaluation results are shown in Table 1.  As shown in the last column ‘Motion Fidelity’ of the Table. 1, the method (w/o AD-Loss) corresponds to learning without AD-Loss,  it only gains a slight improvement compared with the foundation model, while the method (ours) with the AD-Loss achieves improvements with larger gaps. This demonstrates the effectiveness of the effectiveness of AD-Loss.

---

> ### Author Response · Authors · 2023-11-14
> **Response to Reviewer QKrd (4/4)**
>
> **Q4: How many frames can be generated consistently?**
> >Question 2: Do the authors test how many frames can be consistently generated using the proposed method?
>
> **A4:** How many frames can be consistently generated depends on the foundational models, to a large extent. The proposed method and the foundational models we used can consistently generate 16-32 frames. The maximum number of frames that can be consistently generated by existing open-sourced foundational models is 32. If applying the pre-trained temporal interpolation model, the number of frames can be further extended.

---

> ### Author Response · Authors · 2023-11-16
> **Follow-up on Initial Rebuttal Submission**
>
> Dear Reviewer QKrd,
>
> Thank you for your valuable feedback on our submission. We have read your comments carefully and have addressed them in our rebuttal. We would be grateful if you could acknowledge if our responses have addressed your comments. We would also be happy to engage in further discussions if needed. Thank you again for your time and consideration.
>
> Best regards,
>
> Authors of paper 3534.

---

### Official Review · Reviewer_E7V1 · 2023-11-01

**Soundness:** 3 good
**Presentation:** 3 good
**Contribution:** 2 fair
**Rating:** 5
**Confidence:** 5

**Summary:**

This paper proposes MotionDirector, a dual-path LoRA architecture to decouple the learning of appearance and motion within videos for transferring the motion. It also proposes a new loss function for debiasing appearance bias in temporal information. Experiments show that the proposed method can generate diverse videos with desired motion concepts.

**Strengths:**

1. The paper is well-written and easy to follow
2. The idea of dual-path model combining LoRA is intersting,
3. Experimental results show the effectiveness of the proposed method to achieve the transfer of target actions.

**Weaknesses:**

1. In this paper, the LoRA technique is used to decouple the learning of appearance and dynamics in reference videos. Does this method require separate training for each specific set of videos for a particular motion to generate videos? How does the video quality fare beyond the distribution?

2. The authors have designed two branches for learning the appearances and dynamics of videos. In the temporal branch, the authors have included spatial LoRA and share parameters with the spatial branch. Does such inclusion of spatial LoRA in the temporal path interferes with learning temporal information?

3. In Figure 4, video 1 & video 2, and video 2 & video 3 are relatively close, but they do not seem to have much in common (e.g., the three videos do not share the same motion or appearance). On the contrary, video 3 and video 4 are farther apart. Therefore, I am skeptical about the claim that the distance between clusters is determined by the appearance of the videos. It's important to base such claims on statistical results from a larger sample rather than a few videos, as the latter can lead to a higher degree of randomness. I suggest gathering more video data to support the argument.

4. In Figure 4, Part D, the authors mention that it represents the visualization of appearance-debiased latent codes, but it's not clear how it relates to Part C. How does Part D reflect appearance debiasing?

5. In the section "Temporal LoRAs Training" on page 6, why does inserting spatial LoRA into the temporal path allow the temporal LoRA to ignore the appearance information in the training dataset? Why wont it affect spatial LoRA during the training of the temporal path?

6. How was Equation 6 derived? Why was this form of loss function with sampling used to eliminate appearance bias in the temporal information? Epsilon_anchor is used as an anchor, so why is there another epsilon_i later on? What is the purpose of having two anchors?

**Questions:**

see weaknessess

---

> ### Author Response · Authors · 2023-11-14
> **Response to Reviewer E7V1 (1/3)**
>
> Thank you very much for your review! We have prepared a detailed response and hope it satisfactorily addresses your concerns.
>
> **Q1: About the training and quality.**
> > Weakness 1: In this paper, the LoRA technique is used to decouple the learning of appearance and dynamics in reference videos. Does this method require separate training for each specific set of videos for a particular motion to generate videos? How does the video quality fare beyond the distribution?
>
> **A1:** For each set of videos representing a particular motion, we train one MotionDirector to learn it. This is common for customization tasks, such as the DreamBooth [1] for appearance customization. For the video quality, the quality of appearance is similar to the foundation model, while the quality of motions can be improved by MotionDirector, because it learns reasonable motions directly from the reference videos, as shown in Figure 5.
>
> **Q2: About the inclusion of spatial LoRA in the temporal path.**
> > Weakness 2: The authors have designed two branches for learning the appearances and dynamics of videos. In the temporal branch, the authors have included spatial LoRA and share parameters with the spatial branch. Does such inclusion of spatial LoRA in the temporal path interferes with learning temporal information?
>
> **A2:** The parameters of spatial LoRA are only updated in the spatial path to fit the appearances of reference videos. In the temporal path, the spatial LoRA with shared parameters is fixed and will not influence the learning of motion dynamics.
>
> [1] Ruiz, Nataniel, et al. "Dreambooth: Fine-tuning text-to-image diffusion models for subject-driven generation." Proceedings of the IEEE/CVF Conference on Computer Vision and Pattern Recognition. 2023.

---

> ### Author Response · Authors · 2023-11-14
> **Response to Reviewer E7V1 (2/3)**
>
> **Q3: About Figure 4 and its conclusion.**
>
> >Weakness 3:  In Figure 4, video 1 & video 2, and video 2 & video 3 are relatively close, but they do not seem to have much in common (e.g., the three videos do not share the same motion or appearance). On the contrary, video 3 and video 4 are farther apart. Therefore, I am skeptical about the claim that the distance between clusters is determined by the appearance of the videos. It's important to base such claims on statistical results from a larger sample rather than a few videos, as the latter can lead to a higher degree of randomness. I suggest gathering more video data to support the argument.
>
> **A3:** Videos 1-4 are the same as the videos in the first and fourth rows of Fig. 2. To be more specific, video 1 and video 4 are real videos providing the motions that a subject is moving to the left and the right, respectively. Video 2 and video 3 are generated videos with the same appearance which is provided by the static image, as shown in the last row of Fig. 2. Video 2 is animated with the motion of video 1, while video 3 is animated by the motion of video 4. As shown in Fig. 4 (a), the relationships of these four videos are as follows:
>
> * i) video 1 and video 2: sharing similar motions, but with different appearances
>
> * ii) video 2 and video 3: with different motions, but sharing similar appearances
>
> * iii) video 3 and video 4: sharing similar motions, but with different appearances
>
> Take a look at Fig. 4 (c), we can find that the clusters of videos (video 1 & video 2, or video 3 & video 4) that share similar motions have similar internal connectivity structures, but lie in different locations, while the clusters of videos (video 2 & video 3) that share similar appearances lie in closed locations but have different structures. Thus, we have the conclusion that the distance between clusters is primarily affected by the difference in appearance.
>
> The reason for this phenomenon is that the frames from the same video have similar appearances while the motion dynamics among them slightly change their latent codes, so their latent codes lie in relatively close locations. The appearances of frames from different videos are much more different from each other, especially compared to those frames from the same video, reflected in that the distances between clusters are much larger than the inner distances of each cluster. Thus we claim that the distance between clusters is primarily affected by the difference in appearance.
>
> For the statistical analysis, it is hard to visualize the relationships between a large set of latent codes clearly in one figure just like Fig. 4, but we have conducted experiments on 95 videos with 12 different motions in section 4.1. As shown in the last column ''Motion Fidelity'' of the Table. 1, the method (w/o AD-Loss) corresponds to learning without appearance-debiasing operation, it only gains a slight improvement compared with the foundation model, while the method (ours) with the appearance-debiasing operation achieves improvements with larger gaps. This demonstrates that learning the motions directly on the original latent codes, like those in Fig. 4 (c), is much harder than learning the motions on the appearance-debiased latent codes, like those in Fig. 4 (d). Since it is clear in Fig 4. that the appearance-debiasing operation mainly changes the distances between clusters, we conclude that these distances, mainly reflecting the appearance differences, pose a challenge for learning motions.

---

> ### Author Response · Authors · 2023-11-14
> **Response to Reviewer E7V1 (3/3)**
>
> **Q4: About the Part D in Figure 4.**
> >Weakness 4: In Figure 4, Part D, the authors mention that it represents the visualization of appearance-debiased latent codes, but it's not clear how it relates to Part C. How does Part D reflect appearance debiasing?
>
> **A4:** The points in Part D are generated by applying the appearance-debiasing operation to the points in Part C, as formulated in Equation 5. After applying this operation, the bias between different clusters, caused by appearance differences, is removed, while their internal connectivity structures are maintained. Therefore, Part D reflects the result of appearance debiasing. We will add a clearer explanation about how we get Part D from Part C in the paper.
>
> **Q5: About the inclusion of spatial LoRA in the temporal path.**
> >Weakness 5: In the section "Temporal LoRAs Training" on page 6, why does inserting spatial LoRA into the temporal path allow the temporal LoRA to ignore the appearance information in the training dataset? Why wont it affect spatial LoRA during the training of the temporal path?
>
> **A5:** The spatial LoRAs are trained to fit the appearance of videos in the spatial path, and the spatial LoRAs in temporal sharing parameters with them can also fit the appearance. Then the temporal loss is mainly caused by motion dynamics among frames, so the temporal LoRAs can be trained to fit the motions while ignoring the influence of appearances. In the temporal path, only the parameters of temporal LoRAs are updated while the parameters of spatial LoRAs are fixed, so it does not affect the spatial LoRAs.
>
> **Q6: About Equation 6.**
> >Weakness 6: How was Equation 6 derived? Why was this form of loss function with sampling used to eliminate appearance bias in the temporal information? Epsilon_anchor is used as an anchor, so why is there another epsilon_i later on? What is the purpose of having two anchors?
>
> **A6:** Equation 6 is derived from the basic temporal loss function formulated in Equation 1, where the original noise $\epsilon$ is replaced with the appearance-debiased noise $\phi(\epsilon)$, as formulated in Equation 5. The appearance-debiasing operation can eliminate the appearance bias as discussed in the response to the Q3. $\epsilon_i$ represents the noise added to the i-th frame, not an anchor. There is only one anchor $\epsilon_{anchor}$ in Equation 6. We will add a clearer explanation of this in the paper.

---

> ### Author Response · Authors · 2023-11-16
> **Follow-up on Initial Rebuttal Submission**
>
> Dear Reviewer E7V1,
>
> Thank you for your valuable feedback on our submission. We have read your comments carefully and have addressed them in our rebuttal. We would be grateful if you could acknowledge if our responses have addressed your comments. We would also be happy to engage in further discussions if needed. Thank you again for your time and consideration.
>
> Best regards,
>
> Authors of paper 3534.

---

> ### Author Response · Authors · 2023-11-18
> **Response to Reviewer E7V1**
>
> Dear Reviewer E7V1,
>
> Thank you for your valuable feedback on our submission.  We have updated the visualization of relationships of videos 1-4 in Fig. 2. Would you mind telling us if we have addressed your concerns? Thanks very much.
>
> Best regards,
>
> Authors of paper 3534.

---

### Author Response · Authors · 2023-11-14
**Reply to AC and all reviewers**

We thank all the reviewers for their valuable comments. We also appreciate their recognition of the core contributions of our paper and the quality of our results:

1. Contribution to the task of motion customization:  ''The paper identifies the challenge in generalizing customized motions across diverse appearances.'' (Reviewer LtG6)
2. The novelty and effectiveness of the proposed MotionDirector:

     2.1. ''The idea of a dual-path model combining LoRA is interesting.'' (Reviewer E7V1)

     2.2. ''The way that authors decouple the motion and appearance when using LoRA is novel and smart.'' (Reviewer QKrd)

     2.3. ''The proposed appearance-debiased temporal loss sounds reasonable.'' (Reviewer 8uoP)

     2.4. ''The integration of motion in the video appears great, and this effect can be attributed to the decoupling treatment of the temporal module.'' (Reviewer LtG6)

     2.5. ''It proposes a dual-path architecture designed to separate the learning of appearance and motion.'' (Reviewer LtG6)

3. The quality of our results is recognized by all the reviewers.

Below, we respond to the individual concerns of each reviewer, and we are very happy to discuss if you have any further questions.